# Oncogenic Functions and Clinical Significance of Circular RNAs in Colorectal Cancer

**DOI:** 10.3390/cancers13143395

**Published:** 2021-07-06

**Authors:** Maria Radanova, Galya Mihaylova, Neshe Nazifova-Tasinova, Mariya Levkova, Oskan Tasinov, Desislava Ivanova, Zhasmina Mihaylova, Ivan Donev

**Affiliations:** 1Department of Biochemistry, Molecular Medicine and Nutrigenomics, Medical University of Varna, 9000 Varna, Bulgaria; maria.radanova@gmail.com (M.R.); galya.mihaylova@mu-varna.bg (G.M.); neshe.tasinova@mu-varna.bg (N.N.-T.); oskan.tasinov@mu-varna.bg (O.T.); desiplamenova@gmail.com (D.I.); 2Laboratory of Molecular Pathology, University Hospital “St. Marina”, 9000 Varna, Bulgaria; 3Department of Medical Genetics, Molecular Medicine and Nutrigenomics, Medical University of Varna, 9000 Varna, Bulgaria; mariya.levkova@mu-varna.bg; 4Clinic of Medical Oncology, Military Medical Academy, 1000 Sofia, Bulgaria; zhasmina.mihaylova@vma.bg; 5Clinic of Medical Oncology, Hospital Nadezhda, 1000 Sofia, Bulgaria

**Keywords:** circular RNAs, circRNAs, colorectal cancer, oncogenic

## Abstract

**Simple Summary:**

Over the last few years, circular RNAs (circRNAs) are highly anticipated as new diagnostic and prognostic biomarkers. Our review aimed to present new data about circRNAs as targets in oncogenic axes to inhibit cancer activity. The review of circRNAs was made in the context of their oncogenic functions. Special attention was paid to circRNAs with clinical significance and to circRNAs with diagnostic and prognostic potential in colorectal cancer (CRC). Studies that evaluate the differential expression of circRNAs in the circulation between patients with CRC and healthy controls were considered. The review could be useful with its comprehensive information about known cicrRNAs with oncogenic function in CRC and with the questions it raises about the reliability of circRNAs as non-invasive biomarkers.

**Abstract:**

Colorectal cancer (CRC) is ranked as the second most commonly diagnosed disease in females and the third in males worldwide. Therefore, the finding of new more reliable biomarkers for early diagnosis, for prediction of metastasis, and resistance to conventional therapies is an important challenge in overcoming the disease. The current review presents circular RNAs (circRNAs) with their unique features as potential prognostic and diagnostic biomarkers in CRC. The review highlights the mechanism of action and the role of circRNAs with oncogenic functions in the CRC as well as the association between their expression and clinicopathological characteristics of CRC patients. The comprehension of the role of oncogenic circRNAs in CRC pathogenesis is growing rapidly and the next step is using them as suitable new drug targets in the personalized treatment of CRC patients.

## 1. Introduction

Colorectal cancer (CRC) is ranked as the second most commonly diagnosed disease in females and the third in males worldwide [1]. In 2018, CRC was responsible for approximately a million deaths, ordering it in second place for causing cancer-related death [2]. The absence of non-invasive biomarkers for early diagnosis, the high potential of metastasis, and the resistance to conventional therapies determine CRC patients’ poor outcomes. This is why it is essential that CRC initiation and progression mechanisms are clarified, and the research on non-invasive and relatively independent biomarkers for diagnosis, prognosis, and prediction is expanded. In recent years, such biomarkers are searched not only among the circulating proteins and mRNA of cells, but also as cell-free RNA products of different types, including miRNA, linear lncRNAs, and circRNA.

Circular RNAs (circRNAs) have proven to be much more a random by-product of the splicing process [3,4]. Their circular form makes them resistant to exonuclease activity and thus more stable than linear RNAs [5]. Moreover, they have a cell-specific or developmental-stage specific expression pattern [6]. The most important circRNAs’ function is miRNA sponges [7], but despite their place in the group of non-coding RNAs, it was shown that circRNAs could be translated [8,9]. Similar to miRNAs, circRNAs can act as oncogenes and tumor suppressor genes in a concept to cause or to prevent cancer initiation and development [10]. They regulate the tumor microenvironment via the immune system and angiogenesis modulation, extracellular matrix remodeling, and endothelial cell permeability improvement [3,4]. They can be detected not only in tissue biopsy from CRC patients, but also in a liquid one. Based on the literature review, there are many studies that indicate dysregulated circRNAs in CRC tissues and cells, in blood from CRC patients, in secreted from various cells exosomes [11,12,13,14,15]. Moreover, their oncogenic function is explained by a different mechanism like miRNA sponging, sponging or enhancing protein function, peptide translation, cancer-related signaling pathway regulation [11,16].

In the present review, we first briefly summarize discovery, biogenesis, classification of circular RNAs and their function and properties and then present all reported oncogenic circRNAs until mid-March 2021. We pay special attention and highlight the current knowledge on oncogenic circRNAs with clinical significance in cohort studies—circRNAs whose expression levels significantly correlated with clinicopathological features as tumor size, TMN grade, lymph node metastasis, distal metastasis, etc.; circRNAs, which affect chemoradiation resistance; and circRNAs with diagnostic and prognostic potential.

We focus separately on more than 101 circRNAs with oncogenic function in CRC and discuss their particular mechanism of action in the context of their functions. Each circRNA in the current review is presented with all its names to exclude the possibility of one circRNA to being mistaken with another against the background of a growing number of studies for circRNA in the last two years. The effects of silencing of individual oncogenic circRNAs—suppression of cell proliferation, migration and invasion, and promotion of cell cycle arrest and apoptosis—are systematized in a table.

Future studies aim to find new circRNAs as clinical biomarkers and studies to target oncogenic circRNAs associated axis to inhibit cancer activity.

## 2. Discovery, Biogenesis, and Classification of Circular RNAs

Circular RNAs are mentioned for the first time in literature in the 1970s, when Sanger et al., 1976 established that infectious RNAs, inducing disease in higher plants, exist as single-stranded covalently closed circular RNA molecules [17]. Limited attention was paid to these RNAs because of the initial understanding that they are by-products of splicing errors [18] or intermediates, escaped intron lariat debranching [19,20]. With the development of second-generation sequencing techniques and computational analysis, circRNA enigma has started to be revealed. Nowadays, thousands of circRNAs are discovered in humans, and studies of their role in different diseases have increased [6,21,22].

An overview of circRNAs biogenesis is made in Figure 1.

CircRNAs belong to long non-coding RNAs with a transcript length of more than 200 nucleotides [23]. There are different classifications of circRNAs. For example, according to their composition, Cui et al., 2018 [24] described three classes of circRNAs: (1) ecircRNAs, containing only exon sequences. This class accounts for over 80% of discovered circRNAs [25,26,27]; (2) ciRNAs (circular intronic RNAs), having intron sequences with 2′→5′-linked intronic lariats, which are located in the nucleus [28,29]; (3) EIciRNAs, when they include exonic and intronic sequences with 3′→5′-linked, which have nuclear localization [30]. Some authors add a fourth class to this classification (4) tricRNAs, which are actually intronic circRNAs [31,32,33] but generated from pre-tRNA by tRNA splicing enzymes [34]. A slightly different classification divides circRNAs into two main groups based on their biogenesis [35]. In the first group are circRNA, which originate from splicing. The second group consists of circRNAs that are generated by backsplicing—EIciRNAs, ecircRNAs, IcircRNAs.

According to their genomic proximity to a neighboring gene, circRNAs could be divided into five categories: (1) sense or exonic, (2) intronic, (3) antisense, (4) bidirectional or intragenic, (5) intergenic [36]. Exonic and intronic circRNAs are exclusively composed of exons and introns, respectively. Antisense circRNAs are generated when there is an overlapping of one or more exons of the linear transcript on the opposite strand. Intragenic circRNAs are transcribed from the same gene locus of the linear transcript in close genomic proximity but not classified into exonic or intronic. Intergenic circRNAs consist of sequences located in the noncoding region.

Liang et al., 2020 [37] classified circRNAs in nine types, combining the classifications based on their parts and sources: (1) ecircRNAs; (2) EIciRNAs; (3) ciRNAs; (4) intergenic circRNAs (5) tricRNAs; (6) antisense circRNAs; (7) overlapping circRNAs; (8) circRNA rRNAs; (9) intragenic circRNAs.

A recent study reported a novel class of circRNA. Vo et al., 2019 [38] identified the so-called read-through circRNAs, which contain exons that originate from multiple genes, but the authors did not comment or propose any circRNA classification.

## 3. CircRNA Resources Onlines

The growing scientific interest in circRNAs resulted in the creation of a large number of databases. Freely accessible circBase at www.circbase.org (accessed on 13 June 2021) was developed by Glazar P et al., 2014 [39]. The database provides sequence and expression information. The authors make no provision to cover viroids, which were already collected in other resources [40,41]. Circ2Traits is another database reachable at http://gyanxet-beta.com/circdb/ (accessed on 13 June 2021) [42]. Users can find potential human disease-associated circRNAs, download the image of circRNA-miRNA-mRNA-lncRNA interactome network of individual diseases and get information about disease-associated SNPs on a circRNA. circNet (http://syslab5.nchu.edu.tw/CircNet/) (accessed on 13 June 2021) [43]—provides tissue-specific circRNA expression profiles and circRNA-miRNA-gene regulatory networks. It contains the most up-to-date catalogue of circRNAs; provides a thorough expression analysis of both previously reported and novel circRNAs; generates an integrated regulatory network that illustrates the regulation between circRNAs, miRNAs, and genes. CircInteractome (https://circinteractome.nia.nih.gov/) (accessed on 13 June 2021) [44] searches circRNAs name, the genomic position, and best-matching transcripts of the circRNA; retrieves genomic and mature circRNA sequences; searches RBPs binding to a circRNA and to sequences upstream/downstream of the circRNA; identifies RBPs binding to the circRNA junctions, miRNAs targeting a circRNA; designs divergent primers for circRNAs and siRNAs specific to circRNA. CircRNADb (http://202.195.183.4:8000/circrnadb/circRNADb.php) (accessed on 13 June 2021) [45]—contains 32,914 human exonic circRNAs and users can get detailed information related to exon splicing, genome sequence, IRES, open reading frame, protein-coding potential of each circRNA.

**Figure 1 cancers-13-03395-f001:**
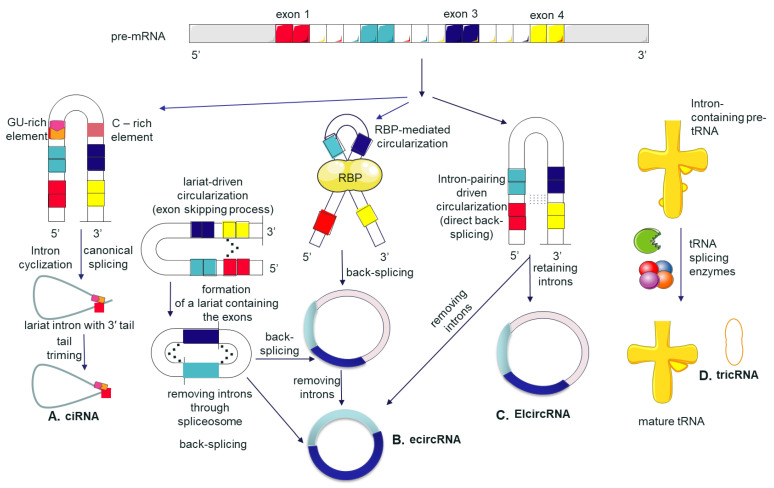
Overview of circRNAs biogenesis: The splicing processes of precursor messenger RNA during maturation may generate different subtypes of circRNAs containing different components from parental genes [46,47]. This splicing process is responsible for the generation of circRNAs in which an upstream acceptor exon links to a downstream donor exon known as back-splicing [26,47,48]. (**A**) Circular intronic RNAs (ciRNAs). They are the result of covalent binding between 7 nucleotide GU-rich element consensus motif near the 5′ splice site and an 11 nucleotide element rich in C close to the branchpoint site [49,50]. (**B**) Exonic circRNA (ecircRNA) RNA-binding protein (RBP)-driven circularization, which depends on sequences specific to RNA-binding proteins, can promote the interaction of the downstream intron and upstream intron, causing the formation of an ecircRNA [32]. Another possibility is when the reverse sequences from introns located on both sides of the pre-mRNA segment are connecting covalently due to complementarity (intron-pairing driven circularization) [51,52]. This possibility is supported by the finding of a tandem repeat of the ALU sequence in the lariat structure [53]. Under lariat-driven circularization, there is a formation of a lariat through exon skipping. Then, the introns are removed through spliceosome to generate ecircRNA [50]. (**C**) Exon-intron circRNAs (EIciRNA). Pairing of quaking (QKI) RNA-binding protein (RBP) may lead to the formation of EIciRNAs, which may form ecircRNA (**B**) by following removal of introns [51,54]. The more the introns include QKI motifs, the higher is the formation of circRNA. This in turn shows that the biogenesis of circRNAs may be regulated by induction or suppression by RBPs depending on the cell needs and conditions [51,55,56]. Similar to ecircRNAs (**B**), EIciRNAs can be produced by intron pairing-driven and lariat-driven circularization also. (**D**) tRNA intronic circular RNAs (tricRNAs). Pre-tRNA transcripts containing eukaryotes are cleaved by a tRNA splicing endonuclease (TSEN) complex, which catalyzes the removal of tRNA introns. This yields two exon halves and an intron. Other single enzyme joins both the exon halves and the intron ends to yield a mature tRNA and a circular intron RNA [57,58].

## 4. Function and Properties of Circular RNAs

Most of the circular RNAs are located in the cytosol [53,59], but circular intronic RNAs (ciRNAs) [49] and exon-intron circRNAs (EIciRNAs) [30] are retained in the nucleus. According to their localization, circRNAs exert different biological functions including regulation of parental genes expression, transcription and modulation of alternative spicing, miRNA sponge affecting expression of respective target genes and proteins, encoding proteins, and RBPs interaction capacity (Figure 2). In CRC context, the frequency of the studies that have detected and reported circRNA roles other than their sponging function is insignificant.

### 4.1. Regulation of Genes Transcription and Alternative Splicing

CircRNAs regulate gene transcription and alternative splicing and it is known that nuclear circRNAs, ciRNAs, and EIciRNAs induce parental genes expression. For example, ciRNAs enhance expression of maternal genes via interaction with RNA Pol II [60] and EIciRNAs can bind U1 spliceosomal RNA, responsible for splicing modification of pre-mRNA, thus regulating RNA pol II [30]. Some circRNAs regulate gene transcription via both RNA polymerase II complex and translation-related machinery [61].

It is known that circularization and splicing compete against each other, enabling circRNAs to contribute to selective splicing regulation [55]. Studies have also reported that exonic circRNAs, which are retained in the nucleus, can form circRNA:DNA hybrid with its parental DNA locus, regulating its splicing. This kind of circRNA:DNA hybrid formation causes the generation of alternatively spliced mRNA by pausing the transcription with exon skipping [62]. According to our knowledge, there are not described oncogenic circRNAs with these classical functions in CRC.

### 4.2. Competing Inhibition of miRNAs

Several long non-coding RNAs and pseudogene RNAs mentioned together as competing endogenous RNAs (ceRNAs) may bind to miRNAs and regulate their inhibitory effect on downstream target genes [63,64]. It was also found that some circRNAs could act as miRNA sponge and suppress or induce the expression of their target genes. The study of Hansen et al., (2013) [65] gives the first strong evidence that circRNAs have many miRNA binding sites and inactivate their inhibitory effect on downstream target mRNAs transcription. Hansen et al. (2013) [65] also first identified the role of circRNAs as “miRNA reservoirs”. According to the authors, circRNAs can release large amounts of miRNAs in certain circumstances to influence the expression of target genes. However, Guo et al., 2014 [6] reported that circRNA class does not contain more miRNA-binding sites than would be by chance [6]. Moreover, the low abundance of most circRNAs in general appears to be a limitation for miRNA sponging hypothesis [66].

In the framework of CRC progression, miRNA sponging is indeed the most widely studied mechanism of circRNAs. There are many representative examples, which are discussed further in the current manuscript.

### 4.3. Interaction with RNA Binding Proteins (RBPs)

CircRNAs can interact with many RNA binding proteins (RBPs), and these interactions are leading to change in the activity of respective proteins [67,68]. circRNA:protein interaction is involved in the competitive inhibition of classical mRNA processing [55], in the binding of RNA pol II-U1 snRNP complex to the DNA, thus stimulating parental gene transcription [30], in the cell cycle arrest [69]. However, similar to miRNA sponging, the low circRNA abundance may be challenging for them to affect protein binding significantly [70]. Regarding colorectal cancer, Karousi et al., 2020 [71] reported two novel circRNAs for both of which various protein-binding sites were detected, and many RBPs were predicted in silico to bind them. Among selected RBPs with more than five binding sites were serine and arginine-rich splicing factors (SRSF), FUS RNA-binding protein, heterogeneous nuclear ribonucleoprotein H1 (HNRNPH1), and others. Although these interactions with RBPs remain in silico predicted for now, they represent another mechanism of circRNAs’ involvement in diverse biological processes, including pathological ones.

### 4.4. Translating Proteins

Does translation of circRNAs occur in cells? This question comes after the fact that the main portion of circRNAs is found in the cytosol [53,59,68]. Although most circRNAs are not associated with ribosomes [6,53], novel studies are pointing out the ability of endogenous circRNAs to be translated and generate proteins [8,72,73,74].

Development of new RNA sequencing technics and their analyses discovered that cyclized RNAs contain open reading frames, which changed the suggestion that they are non-coding RNAs [8,75]. Unlike the mRNA, the structure of circRNAs does not include 5′ 7-methylguanosin cap and 3′ poly-A tail. Therefore, the mechanism of circRNAs translation should be different from the classical cap-dependent one. Initiation of translation and ribosome assembling to circRNAs requires internal ribosome entry sites (IRES), and those lacking them are not translated [73,76]. Recently, several studies have identified that circRNAs can be translational templates in CRC. Bioinformatics analysis predicted circ-BCL2L12-2 to have an open reading frame (ORF) and be subjected to translation [71]. Experimentally verified translational function was reported for circPPP1R12A (hsa_circ_0000423, hsa_circ_001676) and circLgr4 (hsa_circ_02276), both with oncogenic properties [77,78].

Despite the vast presence of circRNAs in the cells, their formation and functions mechanisms are not studied in detail. Future analyses will possibly reveal relevant information about biogenesis and function of circRNAs, giving opportunities for the development of tools in molecular targeting of many socially significant and difficult to treat diseases.

## 5. Oncogenic Functions of circRNAs

Upregulated circRNAs have oncogenic functions of promoting cell proliferation, invasion/migration, metastasis, cell cycle progression, and at the same time decreasing apoptosis rate (Figure 3). Several circRNAs demonstrate more than one oncogenic function as circAGFG1 (hsa_circ_0058514), circCCDC66 (hsa_circ_0001313; hsa_circ_000374), circCSPP1 (hsa_circ_0001806; hsa_circ_001780), circHUWE1 (hsa_circ_0140388), circPRKDC (hsa_circ_0136666), CiRS-7 (CDR1as, hsa_circ_0001946, hsa_circ_105055), etc. The number of studies on the role of individual circRNAs is growing rapidly, new functions and new targets are being revealed, which shows their importance in tumorigenesis.

### 5.1. Impact of circRNAs on Tumor Progression in CRC

Earlier authors divide the progression stages of transformation from normal to cancerous cells into initiation, promotion, and progression to metastasis. In general, colorectal progression is a development from the first to the final stages [79]. It is difficult to define the stages since the progression is an evolutionary process [80]. Often colorectal tumorigenesis is initiated when there are acquired mutations in an APC (adenomatous polyposis coli) gene which handles the Wnt signaling pathway. Activating K-Ras/B-Raf pathway mutations are associated with the growth of a small adenoma to a clinically significant size [80]. CRC development also includes abnormal chromosome segregation, microsatellite instability, hypermethylation of the gene promoter region, loss of function of the p53 gene and others [81]. Key signaling pathways in CRC are Wnt/β-catenin, PI3K/Akt, EGFR/MAPK, Notch, P53, and TGF-β pathways [82,83].

The Wnt/β-catenin signaling pathway is the major one involved in CRC pathogenesis [82]. In the regulation of initiation and progression, the canonical Wnt/β-catenin signaling pathway modulates cancer cells’ apoptosis, proliferation, invasion, migration, angiogenesis, and cell survival [82,84].

PI3K/Akt is another vital signaling pathway related to a variety of cellular activities. As a downstream regulator of PI3K, Akt phosphorylation is associated with tumor cell proliferation and inhibited apoptosis in CRC. The downstream target of Akt is mTOR, which promotes processes like angiogenesis, protein translation, growth, and metabolism [83]. EGFR can trigger the PI3K signaling pathway as well as the MAPK signaling pathway, which can result in the promotion of the proliferation, differentiation, and angiogenesis of CRC cells and inhibition of apoptosis. Wan et al., 2020 [85] reported that 60–80% of colon cancer cases were with overexpressed EGFR. Since various signaling pathways mediate CRC development and progression, it is suggested that they do not function singly but participate in cross-talk or share crucial junctions of interaction.

Several studies have indicated that circRNAs interact with signaling molecules and influence signal-dependent cell functions and regulate cancer development via the presented signaling pathways. The general signaling pathways in CRC that are impact by circRNAs are presented in this section but only in relation with disease progression. Wnt/β-catenin, PI3K/Akt, EGFR/MAPK, Notch, P53, and TGF-β pathways are implicated in the regulation of several other biological processes in tumorigenesis, including invasion/migration, metastasis, angiogenesis, apoptosis, or drug resistance and their mention in other sections of the review could not be avoided.

Additionally, in this section, a place is set aside for displaying of studies which show the role of circRNAs in tumor progression as upregulators of involved in tumorigenesis transcription factors and/or activators of the expression of proven oncogenes.

CiRS-7 (CDR1as, hsa_circ_0001946, hsa_circ_105055) and circHIPK3 (hsa_circ_0000284, hsa_circ_000016, hsa_circ_100782, circPIK3) are involved in more than one signaling pathways in CRC which promote tumor progression [86]. Both circRNAs sequester and reduce the tumor suppressor miR-7, leading to increased expression of miR-7 target proto-oncogenes [87,88,89]. CiRS7 participates in the activation of EGFR/IGF-1R and EGFR/RAF1/MAPK oncogenic pathways [87,88]. In addition, through miR-7-depending upregulation of PRKCB (Protein kinase C beta type), circHIPK3 can activate the NF-κB or Wnt/β-catenin signaling pathway [89]. CircHIPK3 via other miR-7 target proto-oncogenes like as IGF1R and EGFR and could activate PI3K/AKT and MEK/ERK signaling pathways [89]. 

CircRNA participation in the activation of the Wnt/β-catenin signaling pathway is related with upregulation, accumulation, and stabilization of β-catenin in nucleus, and overexpression of genes associated with β-catenin activation. CircCTNNB1 (hsa_circ_0123778) and circAGFG1 (hsa_circ_0058514) increase the transactivation of the transcription factor YY1 (Yin Yang 1), which regulates the expression of genes associated with β-catenin activation [90,91]. CircCTNNB1 interacts with DDX3 (DEAD-box RNA helicase 3) to increase the YY1, while circAGFG1 upregulates YY1 and activates *CTNNB1* transcription by sponging miR-4262 and miR-185-5p [91]. Wei et al., 2021 reported that another circRNAs, circPPP1R12A (hsa_circ_0000423, hsa_circ_001676), also regulates the expression of *CTNNB1* gene for β-catenin via sponging miR-375 [92]. Hsa_circ_100290 (hsa_circ_0013339) increases the expression of *FZD4* gene by inhibiting miR-516b [93]. FZD4 (Frizzled Class Receptor 4) is a receptor of Wnt signaling ligands that induces β-catenin accumulation in the nucleus [93]. CircABCC1 (hsa_circ_001569, hsa_circ_0000677) mediates the entry of β-catenin and binds it into the cell nucleus [94]. Hsa_circ_0005615 (circ5615) also upregulates β-catenin but by miR-149-5p [95]. Authors find that TNKS (tankyrase), a regulator of β-catenin stabilization, is hsa_circ_0005615 downstream and miR-149-5p’s target. CircLgr4 (hsa_circ_02276) has a peptide-coding function and mediates the self-renewal of colorectal cancer stem cells [78]. CircLgr4-derived peptide interacts and activates Lgr4 (leucine-rich repeat-containing G-protein coupled receptor 4), which further promoted the activation of Wnt/β-catenin signaling [78].

There are two studies that do not provide specific mechanism of the Wnt/β-catenin pathway activation by the investigated circRNAs. Chen et al., 2020 speculated that probably circHUWE1 (hsa_circ_0140388) activates the PLAGL2/IGF2/β-catenin signal pathways via miR-486 [96,97]. Jin et al., 2019 found that knockdown of circ_0005075 significantly suppresses CRC progression decreasing β-catenin as well as cyclin D1, c-myc, vimentin, and N-cadherin levels [98].

High expression of circCCDC66 (hsa_circ_0001313, hsa_circ_000374), hsa_circ_0000511 (hsa_circ_002144), hsa_circ_0000392 (hsa_circ_000139), hsa_circ_0104631, and circBANP (hsa_circ_0040824) may regulate PI3K/Akt signaling pathway. Three of them are involved in the activation of PI3K/Akt pathway by sponging miRNAs. Hsa_circ_0000392 sponges miR-193a-5p, which removes the inhibition of PIK3R3; circCCDC66 binds miRNA-510-5p and in this way upregulates its target oncogene AKT2; and hsa_circ_0000511 elevates the expression of prognosis biomarker of CRC LARP1 (La ribonucleoprotein 1, translational regulator) via miR-615-5p [99,100,101]. Hsa_circ_0104631 inhibits the expression of PTEN, and si-circBANP transfection impedes the expression of p-Akt [102,103].

Chen et al., 2020 reported that circRUNX1 (hsa_circ_0002360) acts as a sponge to miR-145-5p, thus blocking it to increase IGF1 (insulin growth factor-1), which has a role in the development and progression of the CRC by PI3K/mTOR signaling pathway [104].

We found four studies for cirRNAs with different mechanisms for stimulation of Hippo/YAP signaling pathway. CircPPP1R12A (hsa_circ_0000423, hsa_circ_001676) encodes a small functional peptide (PPP1R12A-C, circPPP1R12A-73aa), which may have a stimulating effect on this pathway [77]. Hsa_circ_0128846 sponges miR-1184, thus increasing the expression of AJUBA (LIM domain-containing protein ajuba), a protein regulating the signal transmission from the cytoplasm to the nucleus. AJUBA promotes indirectly the oncogenic activity of YAP (YES-associated protein) from the Hippo/YAP signaling pathway [105]. CircFARSA (hsa_circ_0000896, hsa_circ_000263) and hsa_circ_0000512 (hsa_circ_000166) induce activation of the Hippo/YAP signaling pathway as both circRNAs affect LASP1 (LIM and SH3 domain protein 1) expression. CircFARSA sponges miR-330-5p and downstream LASP1 is upregulated, and hsa_circ_0000512 is involved in interaction with the miR-326/LASP1 axis [106,107]. LASP1 facilitates the aggressive CRC cell phenotypes not only by targeting the Hippo/YAP pathway but also activating the PI3K/Akt signaling pathway [108,109].

Chen et al., 2020 [110] described circERBIN (hsa_circ_0001492, hsa_circ_000729) in CRC and revealed that circERBIN/miR-125a-5p/miR-138-5p/4EBP-1 axis is leading to activation of the HIF-1α signaling pathway. Hsa_circ_100859 (hsa_circ_0023064) participates in colon cancer progression via sponging of miR-217 and upregulation of the expression levels of HIF-1α (hypoxia-inducible factor-1α) [111].

Shang et al., 2020 [112] found that circPACRGL (hsa_circ_0069313) acts as a sponge for miR-142-3p/miR-506-3p. A consequence of this, is the increased expression of TGF-β1 (transforming growth factor-β1) and then promoting the transformation of neutrophils from N1 to N2 type, resulting in cancer progression.

Several studies describe circRNAs that promote tumor proliferation and CRC progression by upregulation of the expression of oncogene transcription factors, enzymes with oncogenic properties, and proven oncogenes. Despite the authors trying to explain the specific mechanism underlying circRNAs functions, their roles in regulating the CRC-associated signaling pathways are not covered in detail. CircCAMSAP1 (hsa_circ_0001900) and circMAT2B (hsa_circ_0128498) accelerated CRC progression by upregulation of the expression of oncogene transcription factor E2F1, functioning as a miR-328-5p and miR-610 sponge, respectively [113,114]. CircIFT80 (hsa_circ_0067835) performs its oncogenic properties via miR-1236-3p/HOXB7 axis. HOXB7 (Homeobox protein Hox-B7) is a transcription factor, whose amplifying is closely related to cancer cell proliferation and survival [115,116].

Li et al., 2019 [117] discovered that participation of circFMN2 (has_circ_0005100) in miR-1182/hTERT axis might be implicated in CRC progression. Sponging of miR-1182 leads to upregulated expression of hTERT (human telomerase reverse transcriptase). The increased hTERT activity in CRC allows cancer cells to proliferate unlimited and favors the tumor progression [118]. CircRAE1 (hsa_circ_0060967) sponges miR-338-3p and thereby increasing TYRO3 (protein tyrosine kinase) expression, which promotes CRC cell proliferation and metastasis [119]. The study of Ma et al., 2021 [120] revealed that miR-144 is a target for hsa_circ_0115744. Moreover, the study elucidates that miR-144 directly targets EZH2 (enhancer of zeste homolog 2). EZH2 is a histone-lysine N-methyltransferase enzyme that is significantly upregulated in CRC tissues and cell lines. There is a negative correlation between hsa_circ_0071589 and miR-600 expression levels, and expression of EZH2 is also upregulated by hsa_circ_0071589 [121]. CircFAT1 (hsa_circ_0001461) is also involved in epigenetically modification histone by upregulation of UHRF1 (Ubiquitin-like with PHD and RING finger domains 1) through targeting miR-520b and miR-302c-3p. UHRF1 is involved in tumor growth as an epigenetic regulator of DNA-methylation and histone modification [122].

The target for hsa_circ_0038646 is the upregulation of a gene for GRIK3 (glutamate receptor ionotropic kainate 3) via sponging miR-331-3p [123]. According to the authors, this is the first study that proves the oncogenic functions of GRIK3 and in CRC are still unknown.

One of the mechanisms by which circPRKDC (hsa_circ_0136666) might stimulate CRC progression is via the miR-136/SH2B1 axis, since the adapter protein SH2B1 (Src homology 2 (SH2) and pleckstrin homology (PH) domain-containing protein) acts as an oncogene in different tumors [124]. Wang et al., 2020 described different axis for circPRKDC as a possible mechanism—miR-198/DDR. *DDR1* acts as an oncogene in various tumors and codes receptor tyrosine kinases, which is activated upon binding with collagen [125,126]. Increasing of *DDR1* and *JAG1* (jagged canonical Notch ligand 1) expression through inhibition of tumor suppressor miR-199b-5p is the primary mechanism of action of circNSD2 (has_circ_0008460) [127]. JAG1 is one of the Notch ligands for its activation. The Wnt/β-catenin pathway induces transcriptional activation of JAG1, which makes it the link between Wnt/β-catenin and Notch pathways in CRC [128].

Hsa_circRNA_102209 (hsa_circ_0045890) is involved in the progression of CRC by suppressing miR-761 and upregulating RIN1 (Ras and Rab interactor 1) [129]. RIN1 may affect Ras signaling, and its increased levels promote tumor progression [130].

Pei et al., 2020 [131] explained the oncogenic role of circ_0000218 (hsa_circ_001348) in CRC through its effect on the miR-139-3p/RAB1A axis. Rab1A (Ras-related protein Rab-1A), a member of the RAS oncogene family, is a crucial activator of mTORC1 (mechanistic target of rapamycin complex-1) in CRC [132]. Activated mTORC1 stimulates proliferative signals, thereby favoring the tumor progression.

circSMAD2 (hsa_circ_0000847, hsa_circ_000640) sponges miR-1258 and this results in upregulation of endoplasmic reticulum glycoprotein RPN2 (Ribophorin II) [133]. According to Bi et al., 2018, downregulation of RPN2 may affect cell proliferation, apoptosis, migration, and invasion in CRC by inhibiting activation of the JAK2/STAT3 signaling pathway [134]. However, the specific role of circSMAD2/miR-1258/RPN2 axis for CRC progression is not established.

Hsa_circ_0004277 promotes the proliferation of colorectal cancer cells, acting as a miR-512-5p sponge to upregulate the expression of an oncogene PTMA (Prothymosin alpha) in CRC [135]. The overexpressed PTMA induces the proteasome degradation of tumor suppressor p53 (wild type) through ubiquitination by Mdm2 (E3 ubiquitin-protein ligase), and thus PTMA is free to exert its oncogenic functions. Moreover, probably the co-expression of PTMA and p53 (mutant) is also important for CRC progression [136].

This section of the current review does not cover all circRNAs involved in the progression of the disease but instead presents the studies about circRNAs where they are proven participants in the regulation of CRC signaling pathways or affect the expression of oncogenes. The number of studies on the role of circRNAs in CRC progression increases, and their findings could not be separated from the role of circRNAs in tumor cell proliferation, invasion/migration, and metastasis.

### 5.2. Impact of circRNAs on Invasion/Migration and Metastasis in CRC

Epithelial–mesenchymal transition (EMT) plays a critical role in tumorigenic metastasis and invasion. EMT is a dynamic process in which epithelial cells increase their motility, achieve mesenchymal nature, lose adhesion to the matrix, and enter the bloodstream [137]. This switch between the epithelial to mesenchymal conditions is accompanied by reduced expression of cell junction proteins like E-cadherin and elevated expression of mesenchymal proteins like N-cadherin, fibronectin, and vimentin [138]. In CRC, circRNAs are implicated in activation of EMT and metastasis via changing the expression of transcription factors as ZEB1 (zinc finger E-box binding homeobox 1) or Snail (SNAI1) and Slug (SNAI2), via interaction with matrix mettalopeptidases (MMP2 or MMP14), via regulation of signaling pathways in CRC—Wnt/β-catenin, Hippo/Yap, JAK2/STAT3, and NF-κB signaling pathways. Several studies on the role of specific circRNAs as inducers of EMT, tumor invasion, and metastasis are presented.

CiRS-7 (CDR1as, hsa_circ_0001946, hsa_circ_105055) and circCER (hsa_circ_100876, hsa_circ_0023404) regulate the EMT signaling pathway via sponging miR-135a-5p and miR-516b, respectively [139,140]. Silencing of ciRS7 and circCER leads to upregulation of epithelial-like marker E-cadherin and downregulation of mesenchymal-like markers N-cadherin, vimentin, and Snail. Moreover, according to Zeng et al., 2017, CiRS7 may regulate the pulmonary metastasis in CRC [141]. CircPVT1 (hsa_circ_0001821, hsa_circ_000006) promotes CRC metastasis via inhibition of tumor suppressor miR-145 [142]. However, in all these studies, the molecules acting downstream of the circRNA/miRNA axes do not investigated in contrast to below described circRNAs.

The mechanism of promoting cell migration and invasion of circANKS1B (hsa_circ_0007294) includes increasing Slug levels through the miR-149/*FOXM1* (Forkhead Box M1) axis [143], while circ-KRT6C (hsa_circ_0026416) upregulates Snail expression by the miR-346/*NFIB* (nuclear factor I/B) axis [144,145]. Snail expression is also regulated via hsa_circ_0000512 (hsa_circ_000166) and circCTNNA1 (hsa_circ_0074169) which impair the ERK/Elk-1/Snail pathway. Hsa_circ_0000512 (hsa_circ_000166) is presented earlier in this review as circRNA with an impact on tumor progression in CRC, but it has an effect on migration and invasion by another axis—miR-330-5p/ELK1 (ETS Like-1 protein Elk-1) [146]. According to Zhang et al., 2021, circCTNNA1 upregulates the expression of CXCL5 by targeting miR-363-3p [147]. CXCL5 (C-X-C motif chemokine 5) is a chemokine that mediates neutrophil trafficking, tumor cell migration, and invasion. CXCL5 in CRC promotes liver metastasis by inducing the EMT through activation not only of the mentioned above ERK/Elk-1/Snail pathway but also the AKT/GSK3β/β-catenin pathway [148].

CircABCC1 (hsa_circ_001569, hsa_circ_0000677) as a sponge for miR-145 leads to upregulation of FMNL2 and BAG4 [149]. FMNL2 (formin like protein 2) is included in tumor cell migration and EMT. BAG4 (BAG family molecular chaperone regulator 4), an anti-apoptotic protein, enhances migration and metastasis. Additionally, exosomes carrying circABCC1 may mediate cell stemness and sphere formation and metastasis in CRC [94]. According to Yan et al., 2020 [150], circHIPK3 (hsa_circ_0000284, hsa_circ_000016, hsa_circRNA_100782, circPIK3) might also stimulate FMNL2 interacting with tumor suppressor miR-1207-5p.

He et al., 2018 [151] demonstrated that circACAP2 (hsa_circ_0007331) interacts with miR-21-5p, whose target gene is TIAM1. Team 1 (T-lymphoma invasion and metastasis-inducing protein 1) is upregulated in different cancers, and its level correlates with metastasis [152].

CircCSPP1 (hsa_circ_0001806, hsa_circ_001780) also is involved in the process of EMT and its high levels are associated with higher liver metastasis risk [153]. Wang et al., 2020 [154] explained the role of circCSPP1 in enhancing CRC cell migration and invasion by upregulation of COL1A1 (collagen, type I, alpha 1) expression from circRNA in a miR-193-5p-dependent way.

Hsa_circ_0006174 acts as miR-138-5p sponge and the followed upregulator of MACC1 (Metastasis-associated in colon cancer protein 1) [155]. MACC1 is a transcription factor which, via HGF/cMET axis, is involved in EMT in CRC [156]. Hsa_circ_0020095 positively regulates the transcription factor SOX9 (sex-determining region Y (SRY)-box 9 protein) via sponging miR-487a-3p [157]. SOX9 leads to increased cell invasiveness and metastasis as well as activation of EMT by increasing S100P, a calcium-binding protein from the S100 protein family [158]. We have to note that according to Blache et al., 2019 [159] and Prevostel and Blache, 2017 [160], SOX9 can act as tumor suppressor as well as an oncogene in the intestine epithelium and this effect depends of the dose of SOX9. Hsa_circ_0001178 (hsa_circ_001637) can sponge three miRNAs—miR-382, miR-587, and miR-616, thereby upregulating the expression of their common target transcription factor ZEB1 (zinc finger E-box binding homeobox 1), another way to promote the EMT process [161]. Ren et al., 2020 [161] found two ZEB1 binding motifs in the hsa_circ_0001178 promoter region and showed that this cicrRNA is probably also transcriptionally modulated by ZEB1. Another study suggested a potential diagnostic value not only for hsa_circ_0001178 but also for circ_0000826 (hsa_circ_002032) in colorectal cancer patients with liver metastases [162]. An interesting study for hsa_circ_0000826 showed that hypoxia led to its overexpression, especially in serum [163]. Hsa_circ_0000826 promotes liver metastasis of CRC cells in mice models, and authors suggestede that this circRNAs could be a potential biomarker for the prediction of CRC liver metastasis [164]. Hypoxic derived hsa_circ_0010522 (circ-133) is delivered to normoxic cells and regulates the distribution of E-cadherin across the cell membrane [164]. Authors suggested that hsa_circ_0010522 promotes cancer metastasis via miR-133a. They showed that hsa_circ_0010522 interaction with miR-133a results in positive regulation of GEF-H1 (Rho guanine nucleotide exchange factor) and RhoA. CircZNF609 (hsa_circ_0000615, hsa_circ_000193) promotes cell migration through miR-150, leading to positively regulating Glil transcription factor expression from the Hedgehog pathway [165].

Sponging miR-149 hsa_circ_0011385 (hsa_circ_100146) elevates the expression of HMGA2 (high mobility group AT-Hook 2), which induces EMT [166]. Chen et al., 2019 [167] revealed that overexpression of circNSUN2 (circRNA_103783, hsa_circ_0007380) promotes liver metastasis also through HMGA2. Authors demonstrated that circNSUN2 is involved in the downregulation of E-cadherin, and upregulation of vimentin, in CRC cells but not by miRNA sponging. The role of crcNSUN2 is to stabilize HMGA2 mRNA which is possible via the formation of a triple complex circNSUN2/IGF2BP2/HMGA2 [167]. Other circRNAs that promotes the EMT signaling pathway in CRC without mediation of miRNAs are hsa_circ_101951, circPTK2 (hsa_circ_0005273), and circLONP2 (hsa_circ_0008558). High levels of hsa_circRNA_101951 are associated with increased expression of KIF3A (Kinesin II family member 3A) and mesenchymal marker genes, including N-cadherin, vimentin, and Snail [168]. CircPTK2 promotes in vitro and in vivo EMT of CRC cells by mechanisms involving binding to vimentin [169]. When circPTK2 is targeted by short hairpin RNA (shRNA), tumor metastasizes are suppressed in a patient-derived CRC xenograft mouse model. This confirms its involvement in tumor invasion and metastasis [169]. CircLONP2 promotes the processing of the primary miR-17 through recruiting DGCR8/Drosha complex in RBPs-dependent manner by DDX1 (DEAD-Box Helicase 1). Upregulated and mature miR-17-5p transported to the neighboring cells leads to the spreading of metastasis. Thus, circLONP2 has the regulatory potential to initiate CRC cells’ metastasis [170].

Mechanism of circSMARCC1 (hsa_circ_0003602) to promote tumor is related to the miR-140-3p/MMPs axis. Sponging the miR-140-3p circSMARCC1 prevents the inhibition of metalloproteinases MMP-2 and MMP-9, enhancing cancer cell metastasis activity [171]. Hsa_circ_0007843 also removes the inhibitory effect of miR-518-5p on MMP2 gene expression [172]. Hsa_circ_0053277 accelerates cell proliferation by sponging miR-2467-3p. Matrix metalloproteinase 14 (MMP14) expression is notably upregulated in CRC cells because MMP14 is a downstream target gene of miR-2467-3p [173]. Possible mechanism of circLMNB1 (hsa_circ_0127801) includes upregulation of MMP-2, MMP-9, and N-cadherin expression, which are related to the primary mechanism of cancer cell invasion and metastasis, and EMT [174]. Acting as a miR-101-3p sponge circAPLP2 (hsa_circ_0000372) upregulates its target gene Notch1, thereby stimulating cascades of proliferation and metastasis-related signals with participating of proteins as c-Myc, cyclin D1, and metalloproteinases as MMP-2 and MMP-9 [175]. A study by Liu et al., 2021 [176] revealed another circAPLP2 mechanism promoting CRC metastasis, which involves the miR-495/IL6 axis. Authors acknowledged this axis as a regulatory one for the JAK2/STAT3 signaling pathway activation [176]. CircGLIS2 (hsa_circ_101692) is involved in CRC to enhance the migration phenotype of CRC cells by inhibiting miR-671 and subsequently activation of the NF-κB signaling pathway [177]. Authors concluded that circGLIS2 induces pro-inflammatory chemokine production to enrich the tumor microenvironment with neutrophils. The increased neutrophil/lymphocyte ratio shows that circGLIS2 probably participates in the pro-metastasis environment. The presented large group of studies illustrates that circRNAs play a key role in tumor invasion/migration and metastasis. CircRNAs act via different mechanisms, sometimes affect simultaneously more than one signaling pathways and for part of them further investigation is needed to find their downstream targets.

### 5.3. Impact of circRNAs on Cell Cycle and Apoptosis in CRC

The cell cycle is a highly organized and controlled process. The passing of the cell through the different phases of the cell cycle is regulated mainly by cyclins, cyclin-dependent protein kinases (CDKs), and CDK inhibitors. The deregulation of the cell cycle could explain the tumor proliferation. The critical role for activation of CDKs has dual-specific phosphatases Cdc25 (cell division cycle 25) [178]. Oncogenic circRNAs have a substantial impact on the cell cycle as activators. Some of them directly promote the expression of genes for cell-cycle regulators, while others activate gene expression of proteins, which in their turn modulate various signaling pathways and subsequently regulate the cell cycle.

The mechanism of action of hsa_circ_102958 (hsa_circ_0003854) in CRC is through inhibiting miR-585 to promote expression of the CDC25B gene for a cell-cycle regulator Cdc25B (M-phase inducer phosphatase 2) enzyme [179]. If Cdc25B and Cdc25C (M-phase inducer phosphatase 3) are needful for entry into mitosis as the final effectors of the G2/M phase transition, Cdc25A (M-phase inducer phosphatase 1) has an important role at the G1/S phase transition. Positive regulator of Cdc25A is hsa_circ_0007142 via miR-122-5p [180]. This circRNA sponges also miR-103a-2-5p and miR-455-5p [181,182]. Functional analyses of circCSNK1G1 (hsa_circ_101555, hsa_circ_0001955) point that it competes with miR-597-5p or miR-455-3p to regulate the expression of CDK6 (cyclin-dependent kinase 6) and RPA3 (Replication Protein A3) or MYO6 (Myosin VI) genes, all undisputed oncogenes in CRC [183,184]. The functional role of hsa_circ_000984 (hsa_circ_0001724) is to sponge miR-106b, which also leads to increased expression of CDK6 and inducing the progression of the cell cycle from G1 to S phase [185]. In the study of Ma et al., 2020 the upregulation of Cyclin-D1 gene is validated by hsa_circ_0005615 [95]. The role of circPRTM5 (hsa_circ_0031242) in the cell cycle regulation is exerted via sponging miR-377 and downstream inducing the expression of transcription factor E2F3 [186]. The last one is involved in the expression of cell cycle proteins like cyclin D1 and CDK2 (cyclin-dependent kinase 2). CircABCC1 (hsa_circ_001569, hsa_circ_0000677) upregulates via miR-145, another transcription factor E2F5 from the E2F family, also involved in cell cycle control [149]. The underlying mechanism of circCTNNA1 (hsa_circ_0074169) is sponging of miR-149-5p and upregulating FOXM1 (Forkhead Box M1) expression [187]. FOXM1 is a key protein in the cell cycle progression with expression peaks at S and G2/M phases. Chen et al., 2020 concluded that its upregulation leads to elevating the expression of cyclin B1 and cyclin D1 and suppresses the expression of p21 and p27 in xenograft tumor tissues. Overexpression of circPIP5K1A (hsa_circ_0014130) is related to increased expression of transcription factor AP-1 (activator protein 1) by downregulation of expression of miR-1273a [188]. AP-1 has elevated activity in CRC and might controls cell cycle and cell death through upregulation of cyclin D1, negative modulation of p53, and induction of expression of the anti-apoptotic Bcl genes (Bcl-3, Bim) [189]. CircMDM2 (hsa_circ_0027492) mediates the G1/S transition through decreasing levels and activity of p53, which has a role in inducing cell cycle arrest and apoptosis. The authors claim that circMDM2 has both sponging abilities and less exerts its effect via encoded polypeptide based on CPAT (Coding Potential Assessment Tool) analysis. The mechanism on p53 regulation requires further investigation [190]. Knockdown of hsa_circ_0000069 (hsa_circ_001061, circSTIL) can distinctly suppress cell proliferation, invasion, migration, and lead to the G0/G1 phase arrest of the cell cycle in one CRC cell line [191]. Hsa_circ_0000512 (hsa_circ_000166) is involved in the miR-296-5p/RUNX1 (runt-related transcription factor 1) axis that determines its inhibitory effect on cell cycle arrest at the G0/G1 phase in CRC [192]. Hsa_circ_0055625 promotes the proliferation of CRC through the miR-106b-5p/ITGB8 axis [193]. ITGB8 (integrin β8) regulates the cell cycle by acting upon cell growth and metastasis [194]. CircMBOAT2 (hsa_circ_0007334) acts via sponging miR-519d-3p, which evaluates the expression of TROAP (trophinin-associated protein, tastin) [195,196]. Tastin is a cytosolic protein, and its main role is controlling the assemble of bipolar spindle and the integrity of the centrosome during mitosis. Its overexpression accelerates CRC carcinogenesis [196].

Many studies on the functional mechanism of oncogenic circular RNAs present data from analysis of the impact of circRNAs on CRC cell apoptosis. However, there is scarce evidence of a direct link between circRNAs expression and protein expression related to the regulation of apoptosis. CircCCDC66 (hsa_circ_0001313, hsa_circ_000374) binding miRNA-510-5p may influence colon cancer cell apoptosis via regulating the expression of Bcl-2 family proteins. Authors prove that downregulation of AKT2 by CircCCDC66 activates caspase-9 and increases the apoptosis rate of cancer cells [102]. Inhibition of hsa_circ_0007534 decreases CRC cell proliferation, probably by inducing apoptosis [197]. Zhang et al., 2018 [183] reported a significantly reduced caspase-3 activity and Bcl-2/Bax ratio in CRC cell lines transfected with si-hsa_circ_007534. Bcl-2 and Bax are also significantly changed by circCSNK1G1 (hsa_circ_101555, hsa_circ_0001955) [183]. Silencing of hsa_circ_0004277 significantly increases caspase-3 activity [135]. CircABCB10 (hsa_circ_0008717) negatively regulate miR-326 and upregulates CCL5 (Chemokine (C-C motif) ligand 5) to suppress cell ferroptosis and apoptosis [198]. CircVAPA (hsa_circ_0006990) and hsa_circ_100290 (hsa_circ_0013339) are other circRNAs, which are related to changes in apoptotic rate [93,199]. In contrast to other circRNA studies, cell apoptosis in Li et al., 2019 [199] and Fang et al., 2018 [93] studies is analyzed only by using flow cytometry and it is not proven with detection of anti-apoptotic/proapoptotic proteins or measurement of caspase activity.

### 5.4. Impact of circRNAs on Angiogenesis in CRC

Angiogenesis is the formation of new blood vessels from pre-existing ones. This neovascularization is essential for tumor growth and metastasis. CircRNAs in CRC might regulate the expression of proangiogenic growth factors as EGF (epidermal growth factor) and VEGFA (vascular endothelial growth factor A) via sponging miRNAs. VEGFA is the most significant mediator of tumor angiogenesis, increases vascular permeability and microvascular density. It also activates tumor angiogenesis, vasculogenesis, and metastasis.

Several studies prove the role of circRNAs in increasing the expression of VEGFA in CRC. CircUBAP2 (hsa_circ_0001846, hsa_circ_001335) promotes VEGFA expression but via miR-199a [200]. Another circRNA whose levels positively correlate with the expression of VEGFA is hsa_circ_001971 (hsa_circ_0001060). This effect is explained through the acting of hsa_circ_001971 as a ceRNA for miR-29c-3p [201]. Li et al., 2019 [202] reported that circCCT3 (hsa_circ_0004680) targets miR-613/VEGFA and miR-613/Wnt3 axes, as downregulation of miR-613 leads to high expression of VEGFA and Wnt3. Hsa_circ_0056618 participates in the upregulation of VEGFA and CXCR4, which is possible by reducing miR-206 levels. CXCR-4 (C-X-C chemokine receptor type 4), also known as fusin or CD184, like VEGFA, is closely associated with tumor formation and angiogenesis in different cancers [203].

Angiogenesis-related circRNAs and their mechanism in CRC are little studied in comparison with those in other solid tumors. Further investigations might identify circRNAs as new targets for angiogenesis inhibitors. These agents could help against resistance to anti-angiogenic therapy and might be used in combination with anti-VEGF blockers.

### 5.5. Impact of circRNAs on Chemo- and Radioresistance in CRC

The main therapeutic approaches in CRC are surgical treatment, chemotherapy, radiotherapy (mainly in rectal cancer), and targeted therapy. Chemotherapy is the standard treatment of CRC and radiotherapy and targeted therapy are used in combination with it. The most commonly used chemotherapeutic schemes are based on fluoropyrimidines (intravenous 5-fluoracil (5-FU) or oral capecitabine) in various regimens and combinations—FOLFIRI (5-FU/irinotecan), FOLFOX (leucovorin/5-FU/oxaliplatin) and CAPOX (capecitabine/oxaliplatin) [204]. FOLFOX provides better: response rate (RR), progression-free survival (PFS), and overall survival (OS) in comparison with therapies based solely on fluoropyrimidines [205]. Despite the significant progress of cancer treatment, resistance to therapy remains a significant problem to control the disease. There are several reports on the relationship between circRNAs in CRC and chemo- and radioresistance.

Hsa_circ_0032833 is upregulated in leucovorin, 5-FU, and oxaliplatin (FOLFOX) resistant CRC patient sera and cells and has a regulatory role in chemoresistance in studies of Li and Zheng, 2020 [206] and Hon et al., 2019 [207]. Li and Zheng, 2020 [206] revealed that FOLFOX-resistance mechanism in CRC could be related to hsa_circ_0032833/miR-125-5p/MSI1 axis [206]. Moreover, Hon et al., 2019 [207] suggested that CRC exosomes transfer hsa_circ_0032833 from FOLFOX-resistant cancer cells into FOLFOX-sensitive cells and modulate chemo-resistance in CRC [207].

Exosomes from oxaliplatin resistant cells might also deliver ciRS-122 (hsa_circ_0005963) to sensitive cells, thereby promoting drug resistance [208]. Wang et al., 2020 found that expression levels of ciRS-122 correlate positively with chemoresistance in CRC patients. The oxaliplatin resistance is also associated with high expression of hsa_circ_0079662 in CRC chemoresistant cells [209]. The authors showed the axis hsa_circ_0079662/hsa-mir-324-5p/HOXA9 as a possible mechanism for oxaliplatin resistance. In that light, activation of HOXA9 (Homeobox A9) suggests the involvement of the TNF-α pathway for induction of drug resistance [209].

The levels of circCCDC66 (hsa_circ_0001313, hsa_circ_000374) are increased in CRC cells with resistance to oxaliplatin [210]. Lin et al., 2020 [210] provided evidence that treatment with oxaliplatin induces the high expression of circCCDC66 through PI3KK-mediated DHX9 phosphorylation. Lin et al., 2020 [210] suggested that in the advanced stage of the disease, the elevated circCCDC66 expression, influenced by EMT and/or chemotherapeutic stimulation, may promote cell survival and contribute to the development of chemoresistance and recurrence. This is possible because circCCDC66 impacts the expression of multiple genes simultaneously [211]. Additionally, circCCDC66 is elevated in radio-resistant colon cancer tissues and the colon cancer cells under irradiation exposure [212]. In the study of Wang et al., 2019 [212] circCCDC66 increased cell viability, colony formation rate and decreases caspase-3 activity in colon cancer cells under irradiation by negatively regulating miR-338-3p.

Hsa_circ_0071589 is the next circRNAs with a complex mechanism involving the inhibition of the expression of another miR-526b-3p together with miR-600 [121,213] Hsa_circ_0071589 might regulate cisplatin resistance in CRC targeting oncogene KLF12 (Krüppel-like factor 12) by miR-526b-3p [213]. Hsa_circ_0020095 is another circRNA with investigated effect on cisplatin treatment in CRC cells [157]. Sun et al., 2021 [157] reported that the circ_0020095 resistance towards cisplatin is dose- and time-dependent. Moreover, hsa_circ_0020095 target miR-487a-3p is found to enhance cisplatin sensitivity [157].

Chen et al., 2020 [214] found that circPRKDC (hsa_circ_0136666) levels are raised in 5-FU resistant CRC tissues and cells. In this study, authors claim that high expression of circPRKDC increases β-catenin and c-Myc levels, leading to activation of Wnt/βcatenin pathway in 5-FU resistant CRC cells via miR-375 deletion or FOXM1 (Forkhead box protein M1) overexpression [214]. Hsa_circ_0007031 is proven as upregulated in 5-FU chemoradiation-resistant CRC cells [215,216], and in 5-FU resistant cell lines [217]. Xiong et al., 2017 [215] first bioinformatically predicted signaling pathways involving hsa_circ_0007031, and then Wang et al., 2020 [216] and He et al., 2020 [217] presented functional roles of the cicrRNA and proved its notability as a target for overcoming 5-FU and radiation resistance in CRC. He et al., 2020 [217] described the sponging of miR-133b followed by increasing expression of ABCC5 (ATP-binding cassette subfamily C member 5) as a mechanism involving hsa_circ_0007031 in 5-FU resistance, while Wang et al., 2020, [216] reported the miR-760/DCP1A axis as an underlying mechanism of hsa_circ_0007031 action. Abu et al., 2019 [218] found via microarray that hsa_circ_32883 is upregulated in chemoresistant to 5-FU and oxaliplatin cell line model and is probably involved in drug resistance by sponging miR-130b-5p. Recently, it was shown that miR-130b participates in the modulation of drug resistance via the PI3K/Akt pathway [219]. Nevertheless, these suggestions have to be proven experimentally in CRC.

Xi et al., 2021 [220] claimed that circCSPP1 (hsa_circ_0001806, hsa_circ_001780) is related to DOX (doxorubicin) sensitivity. The underlying mechanism of DOX resistance involves miR-994/FZD7 (frizzled class receptor 7) axis.

Hsa_circ_001680 (hsa_circ_0000598) is involved also in resistance to irinotecan chemotherapy [221]. Hsa_circ_001680 inhibits the expression of miR-340 and upregulates the gene BMI1 (BMI1 polycomb ring finger oncogene). This promotes the cancer stem cell population of CRC cells and induces the chemoresistance [221].

Wang et al., 2021 [222] explained the role of hsa_circ_0067835 in the suppression of CRC cell radiosensitivity by miR-296-5p/IGF1R axis.

The growing number of studies that report circRNAs involvement in therapy resistance determine a reasonable concept of their application as therapy targets. This thought has to be adequately tested, bearing in mind the possible limitations in the implementation of the circRNA treatment strategy. These limitations range from possible side effects of oncogenic circRNA knockdown to the time of molecular profiling and determination of personalized circRNA target treatment during which CRC patients may progress in the disease. In the context of oncogenic circRNA repression as a therapy strategy, there are several tools for manipulation of the expression: (1) circRNA knockdown via antisense oligonucleotides, (2) deletion of circRNA coding gDNA mediated by CRISPR/Cas9, (3) prevention of the backsplicing mediated by CRISPR/Cas9, (4) circRNA degradation mediated by single-guide RNA [223]. These tools are used for a functional characterization of circRNAs, but they need further investigations in order to be developed into circRNA target therapeutic strategies in the future.

### 5.6. Impact of circRNAs on Anti-Tumor Immunity in CRC

PD-1, a co-inhibitory receptor, is highly expressed on activated T cells, B cells, natural killer cells, and myeloid-derived suppressor cells. PD-L1 is a ligand of PD-1. Its expression on healthy tissues is relatively low, but in tumor cells, upregulated expression of PD-L1 is observed [224], which allows tumor cells to avoid anti-tumor immune responses [225].

There are two studies that suggest regulatory functions of circRNAs in anti-tumor immunity in CRC. Zhang et al., 2017 [226] found that expression of hsa_circ_0020397 inhibits miR-138 activity, which leads to elevation of the expression of miR-138 target genes: TERT (telomerase reverse transcriptase) and PD-L1 (programmed death-ligand 1). Perpetuated survival of cancer cells by TERT to suppressed T cell activation by PD-L1, all of which stabilizes telomere length [227]. Additionally, overexpression of ciRS-7 (CDR1as, hsa_circ_0001946, hsa_circ_105055) in CRC cells also leads to increased levels of PD-L1 protein expression at their cell surface [228]. Tanaka et al., 2019 explained this by ciRS-7 induced increasing the expression levels of CMTM4 (CKLF-like MARVEL transmembrane domain containing protein 4) and CMTM6 (CKLF-like MARVEL transmembrane domain containing protein 6) through miRNA-7-independent mechanisms, but likely depends on other undefined factors induced by ciRS-7 expression [228].

Further investigations are required to study thoroughly the anti-tumor immunity-related circRNAs in CRC in the context of their potential predictive role as a biomarker of immune checkpoint therapy.

### 5.7. Impact of circRNAs on Metabolic Pathways in CRC

Cellular metabolism is associated with cancer cell phenotype as specific metabolic pathways can support cell transformation, proliferation, and metastasis. Most of the CRC cells illustrate the “Warburg” metabolic phenotype. It corresponds with upregulated glycolysis, pentose phosphate pathway, nucleotide synthesis, fatty acid synthesis, and downregulated Krebs cycle and mitochondrial oxidative phosphorylation [229]. It seems that switching from aerobic to anaerobic metabolism is a/the driver of tumorigenesis. Furthermore, an acidic tumor microenvironment promotes tumor progression and can induce drug resistance [230]. Moreover, the acquisition of a specific metabolic phenotype allows cancer cells to adapt to environmental stresses during metastasis.

This metabolic reprogramming is ruled by molecular players that drive CRC initiation and progression like Wnt, PI3K, Ras, Myc, and many other proto-oncogenes. For example, Wnt causes the redirection of glucose-derived pyruvate to lactate and lactate secretion [231]. KRAS mutations promote glucose uptake and cell adaptation to glutamine depletion [232,233].

The highlight in recent years is that circRNAs may influence CRC cellular metabolism, thus contributing to metabolic adaptation and disease progression. CircRNAs in CRC are involved in the synthesis, stabilization, and regulation of regulatory enzymes in fatty acid β-oxidation and glycolysis. Li et al., 2019 reported that high levels of circACC1 (hsa_circ_0000759, hsa_circ_001391) are associated with activation of c-Jun and AMPK (AMP-activated protein kinase). Moreover, circACC1 could assemble and stabilize the AMPK holoenzyme. CircACC1 promotes both fatty acid β-oxidation and glycolysis, resulting in changes in the cell’s lipid storage and growth [234]. CircNOX4 (hsa_circ_0023990) is another circRNA involved in CRC metabolism via sponging of the tumor suppressor miR-485-5p, thereby increasing the expression of CKS1B coding CDC28 protein kinase regulatory subunit 1B [235].

The increasing levels of CREBP (cAMP-responsive element-binding protein) have tumor-promoting effects [236]. With that thing considered, Li et al., 2020 [237] found that glycolysis-related enzymes HK2 (hexokinase 2) and LDHA (Lactate dehydrogenase A) are activated by circPRKDC (circ_0136666)/miR-383/CREB1 axis in CRC cells. Hsa_circ_0079993 could also regulate the expression of CREB1 (cAMP response element binding protein 1) by sponging different miRNA- miR-203-3p.1 [238]. Nevertheless, Lu et al., 2019 [238] did not directly associate its increased expression with the cell metabolism. Finally, CircVAPA (hsa_circ_0006990) activates glycolysis and tumor growth via miR-125a/CREB5 axis, and its knockdown mediates a decline in the levels of glucose uptake, lactate production, and ATP in CRC cells [239].

According to Liu et al., 2020 [240], circAPLP2 (hsa_circ_0000372) modulates FOXK1 (forkhead box protein K1) expression by inhibiting miR-485-5p in CRC cells. FOXK1 is a transcription factor that induces aerobic glycolysis, explaining the impact of circAPLP2 on lactate production. Certain circRNAs enhance anaerobic glycolysis as they sponge miRNAs impact on proteins related to glucose uptake and consumption. According to Wang et al., 2020 [208] ciRS-122 (hsa_circ_0005963) sponges miR-122, which normally targets the pyruvate kinase M2 isoform (PKM2). PKM2 is the key regulatory enzyme of glycolysis. It is heterogeneously expressed in CRC cells, high in oxaliplatin-resistant cells and low in sensitive cells, respectively, enhancing glycolysis and ATP production in chemoresistant cells. Zhang et al., 2020 [241] have reported high expression of circDENND4C (has_cicr_0005684) and GLUT1 in CRC tissues and cells. Their upregulation was related to enhanced proliferation, migration, and glycolysis. Authors found that miR-760, which targets GLUT1, is sponged by circDENND4C [241]. CircFAT1 (hsa_circ_0001461) is a sponge for miR-520b and miR-302c-3p. Overexpression of both miR-520b and miR302c-3p is associated with reducing glucose consumption, levels of GLUT1, lactate production, and decreasing activity of HK2 (hexokinase 2), and LDHA (Lactate dehydrogenase A) [122].

There are not many papers linking circRNAs to CRC cell metabolism. However, integrating these studies’ findings gives new insight into circRNAs functioning as a critical player in the acquisition of specific metabolic properties. More in vitro and in vivo are needed to clarify the association between circRNAs and metabolic plasticity.

### 5.8. CircRNAs with Bioinformatically Predicted Impact on CRC Tumorogenesis

Not all studies have implicated circRNA functional analysis for validation of their role in tumorigenesis. Several studies detect circRNAs through microarray or RNA sequencing, select them based on their upregulation and perform in silico analysis to predict putative miRNA targets and miRNA binding sites of circRNAs, to disclose possible signaling pathways, and to construct circRNA-miRNA-mRNA network map. Popular target prediction software used in these circRNA studies are TargetScan and miRanda. Other target prediction programs include DIANA, miRBridge, PicTar, PITA, and rna22. By standing on data for circRNAs targets and binding sites, Cytoscape software can be applied for building circRNA-miRNA-mRNA network map. GO and KEGG analyses reveal the function of upregulated circRNAs.

Here, we describe circRNAs that have been subjected to the mentioned in silico analysis. Xiong et al., 2017 [215] reported hsa_circ_0000504 (hsa_circ_001373), hsa_circ_0007006, and hsa_circ_0007031 as significantly upregulated in 5-FU chemoradiation-resistant CRC cell line. In an attempt to explain this, the authors made in silico analysis that predicts miR-92a-3p and miR-485-5p as direct targets for hsa_circ_0000504. As a possible miRNA targets for hsa_circ_0007006, the authors predict hsa-miR-628-5p and hsa-miR-653-5p and for hsa_circ_0007031—are miR-885-3p and miR-324-5p. *STAT3* is a target gene for miR-485-5p and the authors suppose that miR-485-5p sponging may accelerate chemoradiation resistance in CRC since STAT3 is reported to have a role in chemoradiation resistance development [242]. Interestingly, hsa_circ_0000504, hsa_0007006, and hsa_0007031 are predicted to regulate AKT3 signaling pathway, which could be related to chemoradiotherapy treatment response. Another molecule, predicted to be a subject for hsa_circ_0000504 and hsa_circ_0007031 regulation, is BCL2, which is associated with unresponsiveness towards chemoradiotherapy rectal tumors [243]. By KEGG pathway analysis, other pathways in which these three circRNAs could be involved are the actin-cytoskeleton pathway, focal adhesion signaling, and Wnt/β-catenin signaling pathway.

Two studies contribute with in silico predicted targets that might be part of the underlying mechanism of hsa_circ_0000826 (hsa_circ_002032) in CRC (Table 1) [162,163]. The authors found this circRNA as upregulated in CRC metastatic tissue samples and its diagnostic potential for CRC patients with liver metastasis. Shi et al., 2020 [163] speculated that the hsa_circ_0000826 tumorigenesis promoting mechanism might involve binding to AGO1, 2, 3 (RNA argonaute RISC component 1, 2, 3), EIF4A3 (eukaryotic translation initiation factor 4A), IGF2BP2, 3 (insulin-like growth factor 2 MRNA binding protein 2, 3), and LIN28A (Lin-28 homolog A).

Several other studies described for hsa_circ_0004585, circVAPA (hsa_circ_0006990), hsa_circ_0082182, hsa_circ_0000370 (has_circ_001553), circPNN (hsa_circ_0101802), and hsa_circ_32883, more than one in silico predicted miRNA targets and subsequent these miRNAs’ gene targets, but for hsa_circ_0004831, a theoretically constructed circRNA-miRNA-mRNA regulatory network is presented (Table 1) [199,218,244,245,246,247]. Gene set enrichment analysis shows that hsa_circ_0004831 could participate in the Wnt/β-catenin signaling pathway, EMT, and p53 signaling pathways [247].

This kind of study does not offer conclusions for the functional significance and underlying mechanisms of the upregulated circRNAs. Rather, they provide a good starting point for further investigations. It is possible that circRNAs do not function through their predicted mechanisms, but this is relevant for functional studies since circRNAs are less abundant than miRNAs [70].

### 5.9. CircRNAs as Biomarkers in CRC

The main characteristics of circRNAs are their increased stability and half-life in the tissue, even in paraffin-embedded tissue, in plasma, serum, and saliva [21,38,248,249]. They could be transported by exosomes to the extracellular matrix. Forming of exosomes is considered a mechanism for circRNAs clearance because circRNAs levels are infold higher in extracellular vesicles than in tumor cells [250]. The distinct expression of circRNAs in cancerous and noncancerous, in chemoresistance and chemosensitive tissues, in patients and healthy individuals, or between patients in different stages of the disease emerged circRNAs as potential biomarkers for diagnosis, progression, and chemoresistance in CRC. Finding non-invasive early diagnostic and/or prognostic and/or predictive biomarkers for CRC is an essential challenge in science.

Oncogenic circRNAs presented in the current review are closely associated with clinicopathological characteristics of patients, such as degree of differentiation and size of tumors, TMN stages, lymph node metastasis, and distant metastasis (Table 1). In addition, a number of these circRNAs could be related to the progression of CRC and survival rate—overall survival (OS) or disease-free survival (DFS). Only oncogenic circRNAs, whose diagnostic and prognostic significance has been evaluated by receiver operating characteristic curve (ROC) analysis and by survival analysis such as Kaplan–Meier plots, log-rank tests, and Cox proportional hazards regression model, are indicated as potential diagnostic and/or prognostic biomarker/s in Table 1.

Among the circRNA-focused studies analyzed in this review, a small number is descriptive. These kind of studies investigate associations between circRNAs levels and clinicopathological features of CRC tumors in larger cohorts and improve the understanding of the CRC on a clinical level. These studies evaluate in detail the sensitivity and specificity of particular circRNAs as potential diagnostic and prognostic biomarkers alone and in combination.

There are four studies that present panels of oncogenic circRNAs with potential for non-invasive biomarkers in CRC (Table 2). Ye et al., 2019 [245] showed that hsa_circ_0082182 and hsa_circ_0000370 (has_circ_001553) are upregulated in CRC patients’ plasma. Authors verified the diagnostic potential of hsa_circ_0082182 and hsa_circ_0000370 and when they are combined with a third one hsa_circ_0035445. Ju et al., 2019 [251] reported a high expression of hsa_circ_0122319, hsa_circ_0079480, and hsa_circ_0087391 in CRC tissues and cell lines with high metastatic potential. The authors developed a novel prognostic tool called cirScore. Based on these circRNAs, they successfully classified patients with stage II/III colon cancer into two groups—with low and high risk of disease recurrence [251]. Li et al., 2020 [252] proved the diagnostic value of a panel that consists of hsa_circ_0001900 (circCAMSAP1) and hsa_circ_0001178 (hsa_circ_001637), hsa_circ_0005927. This circRNA panel alone can detect CEA negative CRC patients and differentiate between CRC patients and patients with precancerous lesions. Additionally, the combination of that circRNA panel and CEA levels differentiates more effectively CRC patients from healthy controls than CEA or the panel alone. The fourth study proposes a panel of three downregulated circulating circRNAs—circCCDC66 (hsa_circ_0001313, hsa_circ_000374), circABCC1 (hsa_circ_001569, hsa_circ_0000677), and circSTIL (hsa_circ_0000069, hsa_circ_001061) as a diagnostic biomarker for early-stages CRC and CEA-/CA19-9-negative CRC [253]. The authors demonstrate that low plasma circABCC1 (hsa_circ_0000677, hsa_circ_001569) levels are associated with tumor growth, higher tumor stages, and presence of lymph node and distant metastases. All three circRNAs in the panel are studied independently and are previously proven as upregulated and with oncogenic functions in CRC tissue in other studies [94,102,149,191,210,211,212]. Lin et al., 2019 [253] explained this contradiction with different expression of circRNAs in tissue and plasma and with an uncertain molecular mechanism of secretion of plasma cicrRNAs.

Pan et al., 2019 [254] also demonstrated that the expression of another circNRIP1 (has_circ_0004771) in serum differs from that in CRC cells lines and tissue. CircNRIP1 might distinguish stage I/II CRC patients and all TNM stages CRC patients from healthy individuals with about 80% sensitivity and specificity. This makes circNRIP1 a new potential biomarker for the early diagnosis of CRC patients. Moreover, circNRIP1 in serum might differentiate patients with benign intestinal diseases from stage I/II CRC patients with 81% sensitivity and 74% specificity [254].

On the other hand, in contrast to the descried above studies, several other find simultaneously high expression levels of circRNAs in the blood and in CRC cells and tissues [113,115,117,167,199,244,255].

Finding non-invasive early diagnostic and/or prognostic and/or predictive biomarkers for CRC is an important challenge in science. The constant differential expression of some circRNAs and their significant abundance in serum or plasma indicated them as promising biomarkers. However, at the same time, the established contradictions and differences in the expression of circRNAs in plasma/serum and tumor tissue in CRC patients need to be explained. This illustrates the need for more studies to elucidate the molecular mechanisms of the origin and secretion of circRNAs.

**Table 1 cancers-13-03395-t001:** Upregulated circRNAs involved in CRC progression and associated with CRC patients’ clinicopathological features *.

CircRNA (Alias)	Gene Symbol	Chromosome: Position	Effects of circRNA Silencing	Possible Mechanism	Association with Clinicopathological Features/*biomarker*	Reference
circABCC1 (hsa_circ_0000677 hsa_circ_001569)	*ABCC1*	chr16:16101672- 16162159	proliferation (–) invasion (–)	binds with β-catenin, sponges miR-145	III, IV TNM stages, lymph node, distant metastasis **, poor tumor differentiation	[94,149,253] **
circABCB10 (hsa_circ_0008717)	*ABCB10*	chr1:229665945- 229678118	ferroptosis (+) apoptosis (+) tumor growth (–)	sponges miR-326	-	[198]
circACAP2 (hsa_circ_0007331)	*ACAP2*	chr3:195101737- 195112876	proliferation (–) migration (–) invasion (–)	sponges miR-21-5p	-	[151]
circACC1 (hsa_circ_0000759 hsa_circ_001391)	*ACACA*	chr17:35640167- 35646430	proliferation (–)	probably via promoting AMPK holoenzyme stability and activation	-	[234]
circAGFG1 (hsa_circ_0058514)	*AGFG1*	chr2:228356262- 228389631	proliferation (–) migration (–) invasion (–) stemness (–) apoptosis (+)	sponges miR-4262, miR-185-5p	elevated in CRC patients with liver metastasis	[91]
circANKS1B (hsa_circ_0007294)	*ANKS1B*	chr12:100166699- 100175875	migration (–) invasion (–)	sponges miR-149	lymph node metastasis distance metastasis	[143]
circAPLP2 (hsa_circ_0000372)	*APLP2*	chr11:129979323- 129980556	proliferation (–) migration (–) invasion (–)	sponges miR-101-3p, miR-495, miR-485-5p	*prognostic*	[175,176,240]
circBANP (hsa_circ_0040824)	*BANP*	chr16:88061088- 88098938	proliferation (–)	-	-	[100]
circCAMSAP1 (hsa_circ_0001900)	*CAMSAP1*	chr9:138773478- 138774924	proliferation (–)	sponges miR-328-5p	advanced T stage and clinical stage, better prediction performance than CEA and CA19-9 *prognostic*	[113]
circCCDC66 (hsa_circ_0001313 hsa_circ_000374)	*CCDC66*	chr3:56626997- 56628056	proliferation (–) colony formation (–) migration (–) invasion (–) apoptosis (+) radio-sensitivity (+) chemo resistance (–)	sponges miR-33b, miR-93, miR-185, miR-510-5p, miR-338-3p	*diagnostic,*** * *prognostic*	[102,210,211,212,253] ***
circCCT3 (hsa_circ_0004680)	*CCT3*	chr1:156303337- 156304709	invasion (–) apoptosis (+)	sponges miR-613	advanced TNM stage *prognostic*	[202]
circCER (hsa_circ_100876 hsa_circ_0023404)	*RNF121*	chr11:71668272- 71671937:+	proliferation (–) migration (–) invasion (–) apoptosis (+)	sponges miR-516b	tumor size and differentiation, lymph node and distant metastasis, vascular invasion, clinical T stage, *prognostic*	[140,256]
circCSNK1G1 (hsa_circ_101555 hsa_circ_0001955)	*CSNK1G1*	chr15:64495280- 64508912:-	proliferation (–) apoptosis (+) migration (–) invasion (–)	sponges miR-597-5p, miR-455-3p	≥5cm tumor size, lymph node and distant metastases, III and IV stages, *prognostic*	[183,184]
circCSPP1 (hsa_circ_0001806 hsa_circ_001780)	*CSPP1*	chr8:68018139- 68028357	stemness (–) migration (–) invasion (–) apoptosis (+) chemo sensitivity (+)	sponges miR-193-5p, miR-944	distant metastasis, *prognostic*	[153,154,220]
circCTNNA1 (hsa_circ_0074169)	*CTNNA1*	chr5:138223178- 138260399	proliferation (–) migration (–) invasion (–)	sponges miR-149-5p, miR-363-3p	advanced TNM stage; *prognostic*	[147,187]
circCTNNB1 (hsa_circ_0123778)	*CTNNB1*	chr3:41276543- 41276921	β-catenin activity (–)	binds the Ia domain of DDX3 protein	-	[90]
circDENND4C (hsa_circ_0005684)	*DENND4C*	chr9:19286766- 19305525	proliferation (–) migration (–) glycolysis (–)	sponges miR-760	-	[241]
circERBIN (hsa_circ_0001492 hsa_circ_000729)	*ERBB2IP*	chr5:65284462- 65290692	proliferation (–) migration (–) invasion (–)	sponges of miR-125a-5p, miR-138-5p	-	[110]
circFARSA (hsa_circ_0000896 hsa_circ_000263)	*FARSA*	chr19:13039155- 13039661	proliferation (–) migration (–) invasion (–)	sponges miR-330-5p	*prognostic*	[106]
circFAT1 (hsa_circ_0001461 hsa_circ_000713)	*FAT1*	chr4:187627716- 187630999	proliferation (–) glycolysis (–) apoptosis (+)	sponges miR-520b and miR-302c-3p	-	[122]
circFMN2 (hsa_circ_0005100)	*FMN2*	chr1:240458121- 240497529	proliferation (–) migration (–)	sponges miR-1182	histological grade, lymph nodes metastasis and TNM stage, *diagnostic, * *prognostic*	[117]
circGLIS2 (hsa_circ_101692)	*GLIS2*	-	migration (–) leucocyte recruitment (–)	sponges miR-671	-	[177]
circHIPK3 (hsa_circ_0000284 hsa_circ_100782 hsa_circ_000016 circPIK3)	*HIPK3*	chr11:33307958- 33309057	proliferation (–) migration (–) invasion (–) apoptosis (+)	sponges for miR-7, miR-1207-5p	distant metastasis, TNM stage, liver metastasis, *prognostic*	[89,150]
circHUWE1 (hsa_circ_0140388)	*HUWE1*	chrX:53641494- 53644407	proliferation (–) colony formation (–) migration (–) invasion (–) apoptosis (+)	sponges miR-486	lymphovascular invasion, lymph node and distant metastasis, advanced TNM stage, *prognostic*	[96]
circIFT80 (hsa_circ_0067835)	*IFT80*	chr3:160073800- 160099506	cell growth (–) proliferation (–) colony formation (–) migration (–) invasion (–) apoptosis (+) radiosensitivity (+)	sponges miR-1236-3p, miR-296-5p	tumor size and advanced stage, *prognostic*	[115,222]
circKRT6C (hsa_circ_0026416)	*KRT6C*	chr12:52863194- 52865516	proliferation (–) migration (–) invasion (–)	sponges miR-346	tumor differentiation, TNM stage distant metastasis, lymphovascular and perineural invasion, *diagnostic, * *prognostic*	[144]
circLgr4 (hsa_circ_02276)	*SPTAN1*	chr9: 131369882- 131375764	self-renewal (–) invasion (–)	encodes peptide	*prognostic*	[78]
circLMNB1 (hsa_circ_0127801)	*LMNB1*	chr5:126153227- 126153886	proliferation (–) migration (–) invasion (–) apoptosis (+)	upregulates MMP2 and MMP-9 expression	histological grade, lymph node metastasis and TNM stage, *diagnostic, * *prognostic*	[174]
circLONP2 (hsa_circ_0008558)	*LONP2*	chr16:48311248- 48337216	migration (–) invasion (–)	promotes the processing of primary miR-17	lymph node and distant metastasis, clinical stages (III+IV), *prognostic*	[170]
circMAT2B (hsa_circ_0128498)	*MAT2B*	chr5:162939007- 162943717	proliferation (–)	sponges miR-610	tumor size, lymph node and distant metastasis, TNM stage	[114]
circMBOAT2 (hsa_circ_0007334)	*MBOAT2*	chr2:9083315- 9098771	proliferation (–) migration (–) invasion (–) apoptosis (+)	sponges miR-519d-3p	TNM stage, distant metastasis, lymphovascular invasion, *diagnostic, * *prognostic*	[195]
circ-MDM2 (hsa_circ_0027492)	*MDM2*	chr12:69210591- 69222711	p53 levels (+) E2F targets’ levels (–) growth defects (+) G1-S progression (–)	p53 (unknown mechanism)	-	[190]
circNOX4 (hsa_circ_0023990)	*NOX4*	chr11:89165951- 89185063	proliferation (–) migration (–) invasion (–) glycolysis (–)	sponges miR-485-5p	tumor size, TNM stage, lymph node and distant metastasis, *prognostic*	[235]
circNRIP1 (hsa_circ_0004771)	*NRIP1*	chr21:16386664- 16415895	-	-	*diagnostic*	[254]
circNSD2 (hsa_circ_0008460)	*WHSC1*	chr4:1902352- 1906105	migration (–) invasion (–)	sponges miR-199b-5p	-	[127]
circNSUN2 (circRNA_103783 hsa_circ_0007380)	*NSUN2*	chr5: 6623326- 6625782	migration (–) invasion (–)	interacts with IGF2BP2 and HMGA2 forming a complex	*prognostic*	[167]
circPACRGL (hsa_circ_0069313)	*PACRGL*	chr4:20702035- 20729980	proliferation (–) migration (–) invasion (–)	sponges miR-142-3p, miR-506-3p	-	[112]
circPIP5K1A (hsa_circ_0014130)	*PIP5K1A*	chr1:151206672- 151212515	migration (–) invasion (–) apoptosis (+)	sponges miR-1273a	-	[188]
circPNN (hsa_circ_0101802)	*PNN*	chr14:39648294- 39648666	-	miR6873-3p, miR-6738-3p, miR-6833-3p, let-7i-3p, miR-1301-3p (in silico prediction)	*diagnostic*	[246]
circPPP1R12A (hsa_circ_0000423 hsa_circ_001676)	*PPP1R12A*	chr12:80180153- 80183460	proliferation (–) migration (–) invasion (–) via its encoded product apoptosis (+)	encodes functional peptide, sponges miR-375	*prognostic*	[77,92]
circPRKDC (hsa_circ_0136666)	*PRKDC*	chr8:48715866- 48730122	proliferation (–) migration (–) invasion (–) apoptosis (+) glycolysis (–) colony formation (–)	sponges miR-136, miR-198, miR-375, miR-383	TNM stage, tumor size, lymph node and distant metastasis, *prognostic*	[124,125,214,237]
circPRMT5 (hsa_circ_0031242)	*PRMT5*	chr14:23389732- 23392044	proliferation (–)	sponges miR-377	*prognostic*	[186]
circPTK2 (hsa_circ_0005273)	*PTK2*	chr8:141710989- 141716304	proliferation (–) migration (–) invasion (–)	binds to vimentin protein	*prognostic*	[169]
circPVT1 (hsa_circ_0001821 hsa_circ_000006)	*TCONS*	chr8:128902834- 128903244	migration (–) invasion (–)	sponges miR-145	advanced TNM stage, liver metastasis, *prognostic*	[142]
circRAE1 (hsa_circ_0060967)	*RAE1*	chr20:55931552- 55943868	proliferation (–) migration (–) invasion (–)	sponges miR-338-3p	tumor size, advanced tumor stage, lymph node metastasis	[119]
circRUNX1 (hsa_circ_0002360)	*RUNX1*	chr21:36206706- 36231875	proliferation (–) migration (–) apoptosis (+)	sponges miR-145-5p	lymph node and distant metastasis	[104]
circ_SMAD2 (hsa_circ_0000847 hsa_circ_000640)	*SMAD2*	chr18:45391429- 45423180	tumor growth (–) proliferation (–) invasion (–)	sponges miR-1258	*prognostic*	[133]
circSMARCC1 (hsa_circ_0003602)	*SMARCC1*	chr3:47702783- 47719801	proliferation (–) migration (–) invasion (–)	sponges miR140-3p	-	[171]
circUBAP2 (hsa_circ_0001846 hsa_circ_001335)	*UBAP2*	chr9:33944362- 33956144	proliferation (–) migration (–) invasion (–)	sponges miR-199a	-	[200]
circVAPA (hsa_circ_0006990)	*VAPA*	chr18:9931806- 9937063	proliferation (–) colony formation (–) migration (–) invasion (–) apoptosis (+) glycolysis (–)	sponges miR-19b-1-5p, miR-132-3p, miR-342-3p, miR-101-3p (in silico prediction) miR-101, miR-125a	lymphovascular invasion, lymph node and distant metastasis, TNM stage, *diagnostic*	[199,239]
circZNF609 (hsa_circ_0000615 hsa_circ_000193)	*ZNF609*	chr15:64791491- 64792365	migration (–)	sponges miR-150	histological grade, lymph nodes metastasis, TNM stage, *diagnostic, * *prognostic*	[165]
ciRS-7 (hsa_circ_0001946 hsa_circRNA_105055 CDR1as)	*CDR1*	chrX:139865339- 139866824:+	proliferation (–) migration (–) invasion (–)	sponges miR-7, miR-135a-5p; CMTM4, CMTM6	advanced TNM stages, lymph node metastasis, low histologic grade, larger tumors size, *prognostic*	[86,87,88,139,141,228]
ciRS-122 (hsa_circ_0005963)	*TMEM128*	chr4:4239553- 4248070	glycolysis (–) drug resistance (–)	sponges miR-122	-	[208]
hsa_circ_0000069 (hsa_circ_001061, circSTIL)	*STIL*	chr1:47745912- 47748131	proliferation (–) migration (–) invasion (–)	-	age, II, III TNM stages, *dignostic****	[191,253] ***
hsa_circ_0000218 (hsa_circ_001348)	*DCLRE1C*	chr10:14987103- 15066248	proliferation (–) migration (–) invasion (–)	sponges miR-139-3p	T staging, local lymph node metastasis	[131]
hsa_circ_0000370 (hsa_circ_001553)	*FLI1*	chr11:128628009- 128651918	-	sponges miR-128-3p, miR-502-5p, miR-658 (in silico prediction)	lymph node metastasis, *diagnostic*	[245]
hsa_circ_0000392 (hsa_circ_000139)	*YAF2*	chr12:42604156- 42604482	proliferation (–) migration (–) invasion (–) apoptosis (+)	sponges miR-193a-5p	TNM stage, lymph node and distant metastasis, *diagnostic*	[101]
hsa_circ_0000504 (hsa_circ_001373)	*TUBGCP3*	chr13:113170753- 113181798	-	sponges miR-92a-3p, miR-485-5p (in silico prediction)	-	[215]
hsa_circ_0000511 (hsa_circ_002144)	*RPPH1*	chr14:20811282- 20811431	cell viability (–) proliferation (–) migration (–) invasion (–) apoptosis (+)	sponges miR-615-5p	tumor size, lymph node and distant metastasis, TNM stage, *prognostic*	[103]
hsa_circ_0000512 (hsa_circ_000166)	*RPPH1*	chr14:20811282- 20811436	proliferation (–) apoptosis (+) cell viability (–) colony formation (–) migration (–) invasion (–)	sponges miR-326, miR-330-5p, miR-296-5p	*prognostic*	[107,146,192]
hsa_circ_0000826 (hsa_circ_002032)	*ANKRD12*	chr18:9182379- 9221997	-	interact with RNA binding proteins (in silico prediction), sponges miR-103a-3p, miR-122-5p, miR-1178-3p, miR-1206, miR-107 (in silico prediction)	biomarker for liver metastasis, *diagnostic*	[162,163]
hsa_circ_0001178 (hsa_circ_001637)	*USP25*	chr21:17135209- 17138460	migration (–) invasion (–)	sponges miR-382, miR-587, miR-616	lymph node and distant metastasis, III and IV stages, biomarker for liver metastasis, *diagnostic, * *prognostic*	[161,162]
hsa_circ_0004277	*WDR37*	chr10:1125950- 1126416	proliferation (–) apoptosis (+)	sponges miR-512-5p	tumor size >5cm, III and IV stages, *prognostic*	[135]
hsa_circ_0004585	*KIAA1199*	chr15:81166204- 81212640	-	sponges more than 20 miRNAs (in silico prediction)	tumor size, *diagnostic*	[244]
hsa_circ_0004831	*RSF1*	chr11:77402203- 77404656	-	sponges miR-4326 (in silico prediction)	distant metastasis, differentiation grade, *prognostic*	[247]
hsa_circ_0005075	*EIF4G3*	chr1:21377358- 21415706	proliferation (–) migration (–) invasion (–)	increases expression levels of β-catenin, cyclin D1 and c-myc, thus modulating Wnt/β-catenin pathway	histology differentiation, tumor size, distal metastasis, advanced TNM stage, *prognostic*	[98,257]
hsa_circ_0005615 (circ5615)	*NFATC3*	chr16:68155889- 68157024	proliferation (–)	sponges miR-149-5p	higher T stage, *prognostic*	[95]
hsa_circ_0005927	*VDAC3*	chr8:42259305- 42260979	-	-	tumor size, *diagnostic*	[252]
hsa_circ_0006174	*RAD23B*	chr9:110064315- 110068928	proliferation (–) migration (–) invasion (–) apoptosis (+)	sponges miR-138-5p	TNM stage III, lymph node metastasis, *prognostic*	[155]
hsa_circ_0007006	*DYM*	chr18:46783379- 46808545	-	sponges miR-628-5p, miR-653-5p (in silico prediction)	-	[215]
hsa_circ_0007031	*TUBGCP3*	chr13:113158345- 113181798	proliferation (–) apoptosis (+) sensitivity to 5-FU and radiation (+)	sponges miR-760, miR-133b, miR-885-3p, miR-324-5p *(in silico* prediction)	tumor size, TNM stage, and CEA	[215,216,217]
hsa_circ_0007142	*DOCK1*	chr10:128768965- 128788867	proliferation (–) migration (–) invasion (–) colony formation (–) apoptosis (+)	sponges miR-103a-2-5p, and miR-122-5p, miR-455-5p	poor differentiation, tumor size >5cm, lymph node and distal metastasis, advanced TNM stage	[180,181,182]
hsa_circ_0007534	*DDX42*	chr17:61869771- 61877977	proliferation (–) apoptosis (+)	-	III, IV stages, lymph node and distant metastasis, histological differentiation, *diagnostic * *prognostic*	[255]
hsa_circ_0007843	*ARHGAP32*	chr11:128993340- 129034322	proliferation (–) migration (–) invasion (–) colony formation (–)	sponges miR-518c-5p	-	[172]
hsa_circ_000984 (hsa_circ_0001724)	CDK6	chr7:92462409- 92463134	cell cycle progression (–) proliferation (–) migration (–) invasion (–)	sponges miR-106b	III and IV stages	[185]
hsa_circ_0010522 (circ-133)	*RAP1GAP*	chr1:21939668- 21940582	metastasis (–)	sponges miR133a	-	[164]
hsa_circ_0011385 (circ_100146)	*EIF3I*	chr1:32691771- 32692131	proliferation (–) migration (–) invasion (–) apoptosis (+)	sponges miR-149	-	[166]
hsa_circ_001680 (hsa_circ_0000598)	*B2M*	chr15:45009906- 45009989	proliferation (–) migration (–)	sponges miR-340	-	[221]
hsa_circ_001971 (hsa_circ_0001060)	*UXS1*	chr2:106774513- 106782539	proliferation (–) invasion (–)	sponges miR-29c-3p	TNM stage, *diagnostic, * *prognostic*	[201]
hsa_circ_0020095	*ATRNL1*	chr10:116975454- 117075246	proliferation (–) migration (–) invasion (–) cisplatin resistance (–)	sponges miR-487a-3p	-	[157]
hsa_circ_0020397	*DOCK1*	chr10:128768965- 128926028	cell viability (–) invasion (–) apoptosis (+)	inhibits miR-138 via upregulation of its targets	-	[226]
hsa_circ_0032833	*STON2*	chr14:81837331- 81864924	colony formation (–) migration (–) invasion (–) apoptosis (+) 5-FU, oxaliplatin sensitivity (+)	sponges miR-125-5p	-	[206]
hsa_circ_0038646	*PRKCB*	chr16:23999828- 24166178	proliferation (–) migration (–)	sponges miR-331-3p	III and IV stages	[123]
hsa_circ_0053277	*NRBP1*	chr2:27663993- 27665124	proliferation (–) migration (–) epithelial–mesenchymal transition (EMT) (–)	sponges miR-2467-3p	-	[173]
hsa_circ_0055625	*DUSP2*	chr2:96808907- 96810112	proliferation (–) migration (–) invasion (–)	sponges miR-106b-5p	larger tumor size, TNM stage, histological differentiation, and lymph node metastasis	[193]
hsa_circ_0056618	*SPOPL*	chr2:139259349- 139307864	proliferation (–) migration (–) angiogenesis (–)	sponges miR-206	tumor size, III and IV stage, lymph node and distant metastasis, *prognostic*	[203]
hsa_circ_0071589	*FAT1*	chr4:187517693- 187518946	proliferation (–) migration (–) invasion (–)	sponges miR-600	grade III and IV, lymph node metastasis and advanced clinical stage, *prognostic*	[121]
hsa_circ_0079993	*POLR2J4*	chr7:44043644- 44044934	proliferation (–)	sponges miR-203a-3p.1	III and IV stages, metastasis, *prognostic*	[238]
hsa_circ_0082182	*FAM71F2*	chr7:128317617- 128323309	-	sponges miR-767-3p, miR-609, miR-2682-5p (in silico prediction)	lymph node metastasis	[245]
hsa_circ_0104631	*IREB2*	chr15:78757592- 78778182	proliferation (–) invasion (–)	inhibits PTEN expression and promotes the AKT/mTOR pathway	advanced TNM stage, lymph node and distant metastasis, *prognostic*	[99]
hsa_circ_0115744	*TCONS*	chr21:29818610- 29899681	invasion (–)	sponges miR144	liver metastasis, *diagnostic*	[120]
hsa_circ_0128846	*ZFR*	chr5:32379220- 32420208	proliferation (–) migration (–) invasion (–) apoptosis (+)	sponges miR-1184	-	[105]
hsa_circ_100290 (has_circ_0013339)	*SLC30A7*	chr1:101372407- 101379362	proliferation (–) migration (–) invasion (–) apoptosis (+)	sponges miR-516b	observed in metastatic patients, *prognostic*	[93]
hsa_circ_100859 (hsa_circ_0023064)	*KDM2A*	chr11:66985201- 66986874	proliferation (–) apoptosis (+)	sponges miR-217	TNM stage, histological grade, *KRAS* mutations, *diagnostic, * *prognostic*	[111]
hsa_circ_101951	-	-	proliferation (–) colony formation (–) migration (–) invasion (–) apoptosis (+)	activates the KIF3A-mediated EMT signaling pathway	tumor size, TNM stage, metastasis, *prognostic*	[168]
hsa_circ_102209 (has_circ_0045890)	*PS1*	chr17: 76388556- 76411108	proliferation (–) migration (–) invasion (–) EMT (–) apoptosis (+)	sponges miR-761	histology grade III and IV, liver metastasis, *prognostic*	[129]
hsa_circ_102958 (hsa_circ_0003854)	*PER2*	chr2: 239184383- 239186596	proliferation (–) migration (–) invasion (–)	sponges miR-585	III/IV clinical stages, lymph node metastasis, *prognostic*	[179]
hsa_circ_32883	*EML5*	-	-	sponges miR-130b-5p, miR-367-5p, miR128-3p, miR-501-5p, miR-381-3p (in silico prediction)	-	[218]

* circRNAs’ alias, gene, and chromosome location were extracted from the following online resources: http://bioinformatics.zju.edu.cn/Circ2Disease/circRNAgroup.html, http://www.circbase.org/, http://clingen.igib.res.in/circad/ accessed on 13 June 2021, as well as from the reviewed papers when this was possible. ** low plasma circABCC1 (hsa_circ_0000677, hsa_circ_001569) levels is associated with tumor growth and disease progression [255]. *** diagnostic significance proposed for circCCDC66 (hsa_circ_0001313, hsa_circ_000374) and circSTIL (hsa_circ_0000069, hsa_circ_001061) in plasma [255].

**Table 2 cancers-13-03395-t002:** Panels of circRNAs with potential for non-invasive biomarkers in CRC *.

CircRNA (Alias)	Gene Symbol	Chromosome:	Biomarker	Reference
		Position		
hsa_circ_0082182,	*FAM71F2*	chr7:128317617-128323309	*diagnostic*	[245]
hsa_circ_0000370,	*FLI1*	chr11:128628009-128651918		
hsa_circ_0035445	*ALDH1A2*	chr15:58302846-58306479		
hsa_circ_0122319,	*PLOD2*	chr3:145838898-145842016	*prognostic*	[251]
hsa_circ_0079480,	*CRPPA*	chr7:16298014-16317851		
hsa_circ_0087391	*AGTPBP1*	chr9:88284399-88327481		
hsa_circ_0001900,	*CAMSAP1*	chr9:138773478-138774924	*diagnostic*	[252]
hsa_circ_0001178,	*USP25*	chr21:17135209-17138460		
hsa_circ_0005927	*VDAC3*	chr8:42259305-42260979		
circCCDC66,	*CCDC66*	chr3:56626997-56628056	*diagnostic*	[253]
circABCC1,	*ABCC1*	chr16:16101672-16162159		
circSTIL	*STIL*	chr1:47745912-47748131		

*** circRNAs’ alias, gene and chromosome location were extracted from the following online resources: http://clingen.igib.res.in/circad/, http://www.circbase.org/ accessed on 13 June 2021, http://bioinformatics.zju.edu.cn/Circ2Disease/circRNAgroup.html, as well as from the reviewed papers when this was possible.

## 6. Conclusions

The current review presents circRNAs with their unique features as potential prognostic and diagnostic biomarkers in CRC. However, the diagnostic and prognostic role of a certain circRNA has to be validated after proper statistical analysis of data and confirmation of the results in independent cohorts. Conclusions should not be based only on the differential expression between cancer tissue and healthy tissue in a relatively small number of samples. In our review, we present the mechanism of action and the role of cicrRNAs in the CRC tumorigenesis as well as the association between the expression of oncogenic circRNAs and clinicopathological characteristics of CRC patients as tumor grade, lymph node metastasis, distant metastasis, and patients’ overall survival time. The number of studies on circRNAs is growing rapidly but mainly from Asia. More prospective studies with a larger sample size and from other regions than China have to be performed in order to validate the results for other ethnicities and to improve their clinical applicability in CRC diagnosis and prognosis worldwide.

The clinicians should be aware of cicRNAs’ prognostic role and their future implementation in the diagnostic process. As more data about the oncogenic circRNAs accumulate, they could become the basis of personalized treatment of patients with CRC as suitable new drug targets.

## Figures and Tables

**Figure 2 cancers-13-03395-f002:**
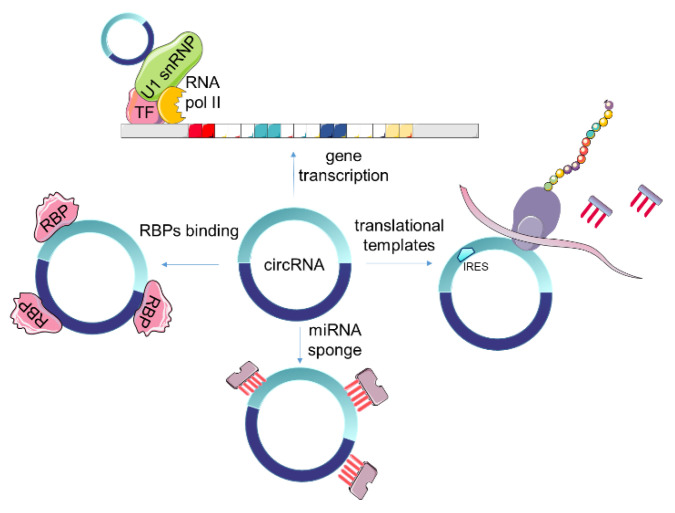
Functions of circRNAs.

**Figure 3 cancers-13-03395-f003:**
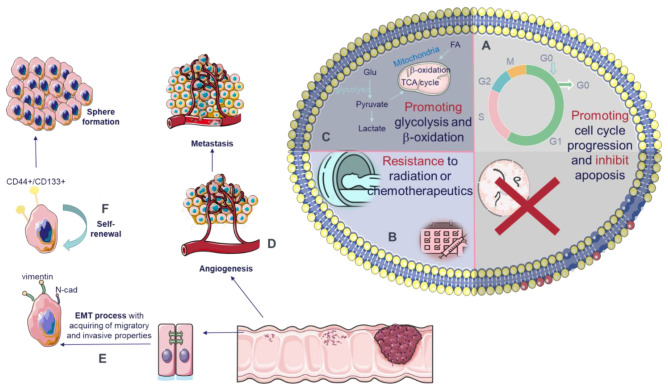
Functions of oncogenic circRNAs in colorectal cancer. (**A**) Oncogenic circRNAs can regulate cell cycle as they allow the passage from one phase into another (Examples: hsa_circ_0000512 (hsa_circ_000166); hsa_circ_102958 (hsa_circ_0003854); hsa_circ_0000069 (hsa_circ_001061, circSTIL)). Part of oncogenic circRNAs is shown to suppress the apoptosis of CRC cells (Examples: circCSNK1G1 (hsa_circ_101555, hsa_circ_0001955); hsa_circ_0007534; circABCB10 (hsa_circ_0008717)). (**B**) For some circRNAs is reported to decrease the sensitivity of the tumor to chemotherapy and radiotherapy (Examples: hsa_circ_0007031; hsa_circ_0020095; circCSPP1 (hsa_circ_0001806, hsa_circ_001780)). (**C**). Several circRNAs are shown to promote glycolytic pathway (Examples: circDENND4C (hsa_circ_0005684); circNOX4 (hsa_circ_0023990); ciRS-122 (hsa_circ_0005963)) and β-oxidation (Example: circACC1 (hsa_circ_0000759, hsa_circ_001391)). (**D**) Some circRNAs are reported to mediate tumor angiogenesis, thus creating an appropriate microenvironment for tumor growth and metastasis (Examples: circUBAP2 (hsa_circ_0001846, hsa_circ_001335); circCCT3 (hsa_circ_0004680)), but most of oncogenic circRNAs are shown to promote metastasis (Examples: hsa_circ_0000512 (hsa_circ_000166); circANKS1B (hsa_circ_0007294); hsa_circ_0005273; hsa_circ_0020095). (**E**) Other circRNAs promote epithelial–mesenchymal transition (EMT), inducing in this way a phenotype with migration and invasive properties (Examples: circCER (hsa_circ_100876, hsa_circ_0023404); circ-KRT6C (hsa_circ_0026416)). (**F**) Some circRNAs can promote the self-renewal of CSCs (colorectal stem cells) (circLgr4 (hsa_circ_02276); hsa_circ_001680 (hsa_circ_0000598)) and their stemness (circAGFG1 (hsa_circ_0058514); circCSPP1 (hsa_circ_0001806, hsa_circ_001780)).

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
