# Peer review of "Oncogenic Functions and Clinical Significance of Circular RNAs in Colorectal Cancer"

_cancers, 2021, doi:10.3390/cancers13143395_

Round 1
Reviewer 1 Report
In the manuscript “Oncogenic function and clinical significance of circular RNAs in colorectal cancer” by Radanova et al. authors review the current literature on the role of circular RNAs in colorectal cancer. This is a comprehensive informative review with a focus on colorectal cancer. However, style issues make some sentences hard to understand and should be corrected. Also, some paragraphs are very long and make the reading difficult. Authors should make an effort to improve these style issues. I have the following suggestions/concerns:
- Section 4.1 is repeated in section 4.2
- Section 4.5 title “Role in protein translation” suggests that circular RNAs play a role in regulating translation but authors mean that circular RNAs can be translated into proteins. The title should be changed for clarity.
- Table 1 focuses on upregulated circular RNAs in colorectal cancer. Is there any data for circular RNAs that might be downregulated compared to normal tissues?
- In the abstract authors mention that circular RNAs might be suitable new drug targets in the personalized treatment of CRC patients, however, this topic is not covered in the review.
- Some paragraphs are very long and difficult to read (example: lines 482-538, lines 712- 773).
- There are some style issues that make some sentences hard to understand and that should be addressed. Some examples:
Line 181: Some circRNAs regulate gene transcription via both RNA polymerase II complex and translation-related [61].
Line 428: The promotion of CRC could be explained by positively regulating of adapter protein SH2B1 expression…
Line 336: circPPP1R12A regulates the expression of CTNNB1 for β-catenin
Table 1 tittle: Upregulated circRNAs involved in CRC progression and associated with CRC patients’ clinicopathological features of upregulated circRNAs in CRC*
Author Response
Reviewer 1
Comment:
In the manuscript “Oncogenic function and clinical significance of circular RNAs in colorectal cancer” by Radanova et al. authors review the current literature on the role of circular RNAs in colorectal cancer. This is a comprehensive informative review with a focus on colorectal cancer. However, style issues make some sentences hard to understand and should be corrected. Also, some paragraphs are very long and make the reading difficult.
Author’s Reply:
We would like to thank you for your thoughtful comments and constructive suggestions, which helped to improve the quality of this manuscript. We tried to identify and correct all sentences that were difficult to understand. We also would like to point out that the two long paragraphs (lines 482-538, lines 712-773) were revised and rewrote. All incorporated changes are presented using the ‘tracked changes’ function in the revised manuscript.
Comment:
Authors should make an effort to improve these style issues. I have the following suggestions/concerns:
- Section 4.1 is repeated in section 4.2
Author’s Reply:
We apologize for our mistake. We deleted section 4.2 and renumbered the other sections.
Comment:
- Section 4.5 title “Role in protein translation” suggests that circular RNAs play a role in regulating translation but authors mean that circular RNAs can be translated into proteins. The title should be changed for clarity.
Author’s Reply:
Thank you for pointing this out! We agree with your opinion that this section title fails to emphasize the content of the text correctly. The title “Role in protein translation” has been changed to “Translating proteins”. In that light, we made a minor change in Figure 2. – “protein translation” was substituted by “translational templates”.
Comment:
- Table 1 focuses on upregulated circular RNAs in colorectal cancer. Is there any data for circular RNAs that might be downregulated compared to normal tissues?
Author’s Reply:
Yes, there are downregulated circRNAs in CRC, and they typically function as tumor suppressors – suppress cell proliferation, invasion and migration and also stimulate cell cycle arrest and apoptosis.
In recent two years, information about circRNAs has increased significantly. Large part of the reviews in the field report the current knowledge on circRNAs biogenesis and mechanisms of action and discuss circRNAs role in the context of many different types of cancer, including CRC. Another part of the reviews is focused on a specific cancer type, but the mentioned upregulated and downregulated circRNAs are used more as examples for summarizing the circRNAs functions in that particular cancer. Moreover, the given examples of circRNAs are often duplicated among the articles, despite the growing number of investigated circRNAs.
The aim of our review was to produce a more systematic, focused, and in-depth report by selecting only upregulated circRNAs with oncogenic functions in CRC. Therefore, not only table 1 but the entire text of the article focuses only on upregulated circRNAs. We are aware that some of the studies are probably missed, but we have put great efforts into synthesizing all of the papers that concern oncogenic circRNAs in the disease. Our study is long and detailed, but useful because together with the more detailed studied circRNAs, with proven oncogenic functions, we also presented weaker studied, but promising circRNAs. We aimed to direct the readers to unexplained problems and opportunities for future perspectives of the understanding of initiation, progression, and functions of circRNAs in CRC.
We could also explain our research team's interest in upregulated circRNAs by obtaining interesting results (unpublished yet) for new circRNAs (hsa_circ_0007915, hsa_circ_0008717, and hsa_circ_0003028) for CRC. We found high levels of these circRNAs in the plasma of CRC patients in comparison with healthy controls and for hsa_circ_0007915, we found that it could significantly distinguish CRC patients in stage III from patients in stage IV.
Comment:
- In the abstract authors mention that circular RNAs might be suitable new drug targets in the personalized treatment of CRC patients, however, this topic is not covered in the review.
Author’s Reply:
Thank you for this important comment! We did not intend to neglect the topic for circRNAs as suitable new drug targets. In fact, this strong conclusion that circRNAs could be effective therapeutic targets is naturally formulated by the growing number of studies that prove the role of circRNAs in tumorigenesis. Moreover, this idea has been developed additionally by several studies, described in the current manuscript, that report circRNAs may cause chemo- and radiotherapy resistance. So far, there are no preliminary studies on that possible circRNA application in clinical practice. The usage of circRNAs as a therapy option for CRC or any other cancer is still in the area of future perspectives. That is why we did not set aside such a section. We decided to focus our review on something already proven and known as the role of circRNAs and their mechanism of action in CRC.
However, in response to your remark, we decided to add a small paragraph as comment in section 5.5. Impact of circRNAs on chemo- and radioresistance in CRC (line 896 to line 908 in revised manuscript).
Comment:
- Some paragraphs are very long and difficult to read (example: lines 482-538, lines 712- 773).
Author’s Reply:
Thank you for your comment. We appreciate your opinion and agree that the paragraphs - lines 482-538 in section 5.2. “Impact of circRNAs on invasion/migration and metastasis in CRC” and lines 712-773 in section 5.5. “Impact of circRNAs on chemo- and radioresistance in CRC” are very long and difficult to read and it would be better to correct them. Therefore, we rewrote and edited these sections. These major changes are illustrated with the help of ‘track change mode’ in the revised manuscript.
Comment:
- There are some style issues that make some sentences hard to understand and that should be addressed. Some examples:
Line 181: Some circRNAs regulate gene transcription via both RNA polymerase II complex and translation-related [61].
Line 428: The promotion of CRC could be explained by positively regulating of adapter protein SH2B1 expression…
Line 336: circPPP1R12A regulates the expression of CTNNB1 for β-catenin
Table 1 tittle: Upregulated circRNAs involved in CRC progression and associated with CRC patients’ clinicopathological features of upregulated circRNAs in CRC*
Author’s Reply:
Thank you for your comments. Your recommendations were considered and accepted and sentences on line 181, line 428, line 336, and Table 1 tittle were corrected as follows:
Line 181 (in revised manuscript line 180):
Some circRNAs regulate gene transcription via both RNA polymerase II complex and translation-related machinery [61].
Line 428 (in revised manuscript lines 439 - 442):
One of the mechanisms by which circPRKDC (hsa_circ_0136666) might stimulate CRC progression is via the miR-136/SH2B1 axis, since the adapter protein SH2B1 (Src homology 2 (SH2) and pleckstrin homology (PH) domain-containing protein) acts as an oncogene in different tumors [125].
Line 336 (in revised manuscript lines 329 - 332):
Wei et al., 2021 report that another circRNAs, circPPP1R12A (hsa_circ_0000423, hsa_circ_001676), also regulates the expression of CTNNB1 gene for β-catenin via sponging miR-375 [93].
Table 1 tittle:
Table 1. Upregulated circRNAs involved in CRC progression and associated with CRC patients’ clinicopathological features*.
We are hoping that we answered your questions and comments and the made corrections have improved the quality of the manuscript.

Reviewer 2 Report
This review article by Radanova, M et al. summarizes the origins and functions of circular RNAs, especially in relation to colorectal cancer. I admire the effort of reading, outlining, and summarizing a wide range of papers, but it is too long and describes various phenomena in parallel, making it difficult to understand the interrelationships. In addition, there are several excellent reviews on the origins and functions of circular RNAs, and there is not enough content to go beyond these reviews. Since the article on colorectal cancer and circular RNAs is detailed and novel, I recommend that the authors revise the article to focus on it.
The reader may be confused because phenomena for which there is sufficient evidence are described mixed with speculations for which there is still insufficient evidence. A strict distinction must be made between them. Also, multiple descriptions of the same phenomenon appear in different parts of the manuscript. For example, descriptions related to Wnt signaling are found in sections 5.1, 5.2, 5.4, 5.7 and 5.8, and 5-FU resistant are in line712 and 750. These should be discussed together. Table 1 is in more than 10 pages long and difficult to read, why not make it an Excel file and use it as supplemental data?
Author Response
Reviewer 2
Comment:
This review article by Radanova, M et al. summarizes the origins and functions of circular RNAs, especially in relation to colorectal cancer. I admire the effort of reading, outlining, and summarizing a wide range of papers, but it is too long and describes various phenomena in parallel, making it difficult to understand the interrelationships. In addition, there are several excellent reviews on the origins and functions of circular RNAs, and there is not enough content to go beyond these reviews. Since the article on colorectal cancer and circular RNAs is detailed and novel, I recommend that the authors revise the article to focus on it.
The reader may be confused because phenomena for which there is sufficient evidence are described mixed with speculations for which there is still insufficient evidence. A strict distinction must be made between them. Also, multiple descriptions of the same phenomenon appear in different parts of the manuscript. For example, descriptions related to Wnt signaling are found in sections 5.1, 5.2, 5.4, 5.7 and 5.8, and 5-FU resistant are in line712 and 750. These should be discussed together. Table 1 is in more than 10 pages long and difficult to read, why not make it an Excel file and use it as supplemental data?
Author’s Reply:
We appreciate your feedback!
We agree with you that there is not enough content in our review to go beyond other reviews on the circRNAs in CRC regarding their origin and general functions. We have to say this was not our goal since the description of circRNAs biogenesis and functions is approved as a mandatory and repeatable part of similar to our review papers. In fact, our goal was to go beyond other review papers in terms of section 5. “Oncogenic functions of circRNAs”, the main part of the current manuscript.
The aim of our manuscript was to present the roles of upregulated circRNAs in the pathogenesis of CRC. We tried to divide them into different groups regarding their oncogenic functions to compare them in the current group and to explain their relationships with downstream targets. If we have understood you correctly, we have to say that in the separated subsections is impossible not to describe various phenomena in parallel, because the different circRNAs use a different mechanism to affect the same oncogenic function. On the other hand, some circRNAs could be involved in different interactions and often have more than one oncogenic function.
Furthermore, the key signal pathways in CRC as EGFR/MAPK, Notch, PI3K, TGF-β, and Wnt signaling pathway are implicated in different oncogenic processes such as cell proliferation, differentiation, invasion, migration, angiogenesis, apoptosis, and survival. That is why the example you gave about Wnt signaling pathway in your comment is found in more than one subsections of the manuscript. We inserted this explanation between line 291 and line 296.
We understand that the text is difficult to read, but at the same time, it is suitable as it contains a set of information with clinically meaningful benefit for oncogenic circRNAs in CRC. With our review, we want to target both readers that are interested in the comprehensive study and readers that are only interested in particular subsections. The current form of the revised manuscript serves both of these readers. Therefore, we would like to keep the manuscript mainly in its current revised form. However, as you had suggested we tried to remove all speculations for which there is still insufficient evidence, as shown in the revised manuscript. We also tried to identify and correct all sentences that were difficult to understand, to revised and rewrote the very long paragraphs. All incorporated changes are presented using the ‘tracked changes’ function in the revised manuscript.
Regarding to your suggestion about Table 1, please, allow us to not agree with your proposal to make it in an Excel file and use it as supplemental data. In Table 1 we summarized important information about functional effects in tumorigenesis after circRNA silencing. Table 1 presents the established association of upregulated circRNAs with clinicopathological features of CRC patients, which we do not discuss in the text. We underlined these circRNAs with proven diagnostic and prognostic potential. All upregulated circRNAs are arranged alphabetically and in ascending ID numbers. This allows readers easy to find information for a single circRNA and all references about it. That is why we claim that Table 1 is an integral part of the manuscript. In the revised manuscript, we tried to reduce the size of Table 1 by deleting full names of circRNAs coding genes since this information is readily available for everyone. We hope that after formatting the table according to the journal’s requirements, if this manuscript is accepted for publication, the size will be reduced additionally.
We are very thankful for your observations and comments. We are hoping that we have understood your comments and our answers are acceptable.

Round 2
Reviewer 1 Report
The authors have addressed most concerns.
Reviewer 2 Report
The authors did not take my advice that it was too long and should be focused on CRC or that the circRNA table should be prepared in excel table as a supplemental file, but they rewrote a ton of text and I felt it was much better each way.
The manuscript is still too long to be accepted by a normal printed journal, but since this journal is open access, I will leave it to the editor's discretion.